# Measuring Fine-Grained Relatedness in Multitask Learning via Data Attribution

**Yiwen Tu**                                                                                          *y2tu@ucsd.edu*
*University of California, San Diego*

**Ziqi Liu**                                                                                     *ziqiliu2@andrew.cmu.edu*
*Carnegie Mellon University*

**Jiaqi W. Ma**                                                                                      *jiaqima@illinois.edu*
*University of Illinois, Urbana-Champaign*

**Weijing Tang**                                                                                  *weijingt@andrew.cmu.edu*
*Carnegie Mellon University*

**Reviewed on OpenReview:** *https://openreview.net/forum?id=zIDGm96xwg*

## Abstract

Measuring task relatedness and mitigating negative transfer remain a critical open challenge in Multitask Learning (MTL). This work extends data attribution—which quantifies the influence of individual training data points on model predictions—to MTL setting for measuring task relatedness. We propose the MultiTask Influence Function (MTIF), a method that adapts influence functions to MTL models with hard or soft parameter sharing. Compared to conventional task relatedness measurements, MTIF provides a fine-grained, instance-level relatedness measure beyond the entire-task level. This fine-grained relatedness measure enables a data selection strategy to effectively mitigate negative transfer in MTL. Through extensive experiments, we demonstrate that the proposed MTIF efficiently and accurately approximates the performance of models trained on data subsets. Moreover, the data selection strategy enabled by MTIF consistently improves model performance in MTL. Our work establishes a novel connection between data attribution and MTL, offering an efficient and fine-grained solution for measuring task relatedness and enhancing MTL models.

## 1 Introduction

Multitask learning (MTL) leverages shared structures by jointly training tasks to enhance generalization and improve prediction accuracy (Caruana, 1997). This paradigm has demonstrated its effectiveness across a range of domains, including computer vision (Zamir et al., 2018), natural language processing (Hashimoto et al., 2017), speech processing (Huang et al., 2015), and recommender systems (Ma et al., 2018). However, when tasks are only weakly related or have conflicting objectives, MTL can degrade performance—a phenomenon known as *negative transfer* (Zamir et al., 2018; Standley et al., 2020). To address this challenge, a central focus in the MTL literature has been modeling and measuring the relatedness among tasks (Zhang & Yeung, 2010; Standley et al., 2020; Worsham & Kalita, 2020; Zhang et al., 2023b).

A straightforward—and arguably gold-standard—approach for measuring task relatedness is to train models under every subset of task combinations, and evaluate the model performance for each combination. However, this approach quickly becomes computationally infeasible as the number of tasks grows (Fifty et al., 2021). Inspired by recent advances in data attribution methods (Koh & Liang, 2017; Park et al., 2023), which aim to efficiently predict the performance of models retrained on data subsets but without actual retraining (Park et al., 2023), we propose to adapt data attribution methods for MTL models as an efficient way to estimate the relatedness among tasks.

To this end, we introduce the *MultiTask Influence Function* (**MTIF**), a data-attribution method tailored for multitask learning. MTIF adapts the influence functions (Koh & Liang, 2017) to MTL models with either hard or soft parameter sharing, providing a first-order approximation of the model performance when removing certain data points from a specific task without retraining the model. The proposed method allows us to efficiently quantify how each sample in a source task influences the performance of a target task.

In comparison to most conventional approaches that measure task relatedness at the entire-task level (Fifty et al., 2021; Wang et al., 2024), the proposed MTIF naturally enjoys a more fine-grained, *instance-level* measurement of task relatedness. As evidenced by recent transfer learning and domain adaptation studies (Lv et al., 2024; Yi et al., 2020; Zhang et al., 2023a), the contribution of different individual examples from a source task to a target task can vary widely: some examples improve target task performance, others have little effect, and some may lead to negative transfer. With the instance-level relatedness measurement, MTIF enables a novel approach to mitigate the negative transfer in MTL through *data selection*.

We validate the effectiveness of MTIF through two sets of complementary experiments. First, on smaller synthetic and HAR (Anguita et al., 2013) datasets where we can afford measuring the gold-standard retraining performance, we show that MTIF's instance-level influence scores correlate almost perfectly with brute-force leave-one-out retraining, and that the task-level relatedness measurements induced by MTIF similarly align with brute-force leave-one-task-out retraining. Second, on large-scale image benchmarks including CelebA (Liu et al., 2015), Office-31 (Saenko et al., 2010), and Office-Home (Venkateswara et al., 2017), we apply MTIF to improve MTL performance through data selection. The proposed method achieves consistent accuracy improvements over state-of-the-art MTL methods at comparable computational cost.

Finally, we summarize our contributions as follows.

- We propose MTIF, which introduces the idea of data attribution into MTL, leading to a fine-grained instance-level relatedness measurement.

- We apply the proposed MTIF to mitigate negative transfer in MTL through data selection.

- We conduct extensive experiments to validate the proposed method in terms of both the approximation accuracy of the influence scores and the effectiveness in improving MTL performance.

## 2 Related Work

### 2.1 Task Relatedness in Multitask Learning

As a central problem in MTL, there has been a rich literature measuring and modeling task relatedness. Existing literature can be roughly divided into three categories, as detailed below. At a high level, most existing works treat task relatedness at the entire-task level, while our proposed MTIF, which is a data-attribution-based approach, naturally measures instance-level relatedness. Moreover, the data selection strategy enabled by the proposed MTIF is orthogonal to many existing MTL methods and could be used in combination with other methods.

**Direct Measurement of Task Relatedness.** Standley et al. (2020) introduced a task grouping framework by exhaustively retraining task combinations to measure inter-task relatedness. To scale this approach, most methods now fall into two categories. The first infers relatedness on-the-fly during training, either by tracking per-task losses or by comparing gradient directions across tasks (Fifty et al., 2021; Wang et al., 2024) (Jeong & Yoon, 2025). While these measures are computationally efficient, they depend heavily on the specific training trajectory, which can limit interpretability. The second category leverages auxiliary techniques—such as task embeddings (Achille et al., 2019), surrogate models (Li et al., 2023), meta-learning frameworks (Song et al., 2022), or information-theoretic metrics like pointwise $\mathcal{V}$-usable information (Li et al., 2024)—but typically incurs additional fine-tuning or retraining overhead. Although task similarity metrics from transfer learning have been explored (Zamir et al., 2018; Achille et al., 2021; Dwivedi & Roig, 2019; Zhuang et al., 2021; Achille et al., 2019), Standley et al. (2020) demonstrated that these do not readily generalize to the MTL setting.

**Optimization Techniques Exploring Task Relatedness.** A complementary line of work focuses on designing MTL optimization algorithms that explicitly account for inter-task relationships. One approach modifies per-task gradients to mitigate negative transfer (Yu et al., 2020; Wang et al., 2021; Liu et al., 2021a;b; Chen et al., 2020; Peng et al., 2024). Another adapts task loss weightings to balance contributions or emphasize critical tasks (Chen et al., 2018b; Liu et al., 2019; Guo et al., 2018; Kendall et al., 2018; Lin et al., 2022; He et al., 2024). Additional methods, such as adaptive robust MTL (Duan & Wang, 2023), dual-balancing MTL (Lin et al., 2023), smooth Tchebycheff scalarization (Lin et al., 2024), and multi-task distillation (Meng et al., 2021), do not fit cleanly into these categories but share the goal of harmonizing task interactions. These optimization strategies are orthogonal to our data-selection approach and could be combined with MTIF for further gains.

**Other Approaches to Task Relatedness.** Several works mitigate negative transfer via specialized MTL architectures. Examples include Multi-gate Mixture-of-Experts (Ma et al., 2018), Generalized Block-Diagonal Structural Pursuit (Yang et al., 2019), and Feature Decomposition Network (Zhou et al., 2023a). These architectural innovations are complementary to our method and illustrate alternative means of capturing task relatedness. Furthermore, approaches based on adversarial alignment (Tzeng et al., 2017; Ganin et al., 2016) and distribution matching (e.g., MMD) (Gretton et al., 2008; Long et al., 2015) aim to improve transfer by aligning source and target representations under distribution shift. Our framework is orthogonal to these optimization-based strategies and can be applied in tandem, for instance, to evaluate cross-task effects or to further improve training via instance-level data selection.

## 2.2 Data Attribution

Data attribution methods quantify the influence of individual training data points on model performance. These methods can be broadly categorized into retraining-based and gradient-based approaches (Hammoudeh & Lowd, 2024). Retraining-based methods (Ghorbani & Zou, 2019; Jia et al., 2019; Kwon & Zou, 2022; Wang & Jia, 2023; Ilyas et al., 2022) require retraining the model multiple times on different subsets of the training data. Retraining-based methods are usually computationally expensive due to the repeated retraining. Gradient-based methods (Koh & Liang, 2017; Guo et al., 2021; Barshan et al., 2020; Schioppa et al., 2022; Kwon et al., 2024; Yeh et al., 2018; Pruthi et al., 2020; Park et al., 2023) instead rely on the (higher-order) gradient information of the original model to estimate the data influence, which are more efficient. Many gradient-based methods, including recent extensions to LLMs (Xia et al., 2024; Wang et al., 2025; Dai et al., 2025), can be viewed as variants of influence function-based data attribution methods (Koh & Liang, 2017). In this paper, we establish a novel connection between data attribution and MTL, leveraging data attribution to measure fine-grained relatedness among tasks and to mitigate negative transfer in MTL. Methodologically, the proposed MTIF is an extension of influence functions to the MTL settings.

In parallel, recent work has explored data curation and coreset selection using proxy criteria beyond attribution scores (Abbas et al., 2024; Moser et al., 2026). These include geometric coverage or boundary reconstruction (Yang et al., 2024), training dynamics and uncertainty signals (He et al., 2023; Cho et al., 2025), information maximization (Tan et al., 2025), and mixture-level optimization policies (Ye et al., 2025; Gu et al., 2025). These directions utilize alternative selection criteria that are complementary to our attribution-based methodology, and could be combined with MTIF to support principled data curation in multitask learning.

## 3 Influence Function for Multitask Data Attribution

We tackle the problem of task relatedness from a data-centric perspective: by quantifying how individual training data from one task contribute to the performance of another, the *instance-level* granularity of which offers finer-grained insights into inter-task interactions. In this section, we develop an IF-based data attribution framework for MTL that builds on the leave-one-out principle. We begin by introducing the general MTL setup and common parameter-sharing schemes.

### 3.1 Problem Setup for Multitask Learning

MTL aims to solve multiple tasks simultaneously by leveraging shared structures. This is especially beneficial when tasks are related or when data for individual tasks is limited. The common approach in MTL to facilitate information sharing across tasks is through either soft or hard parameter sharing (Ruder, 2017). In soft parameter sharing, regularization is applied to the task-specific parameters to encourage them to be similar across tasks (Xue et al., 2007; Duong et al., 2015). In contrast, hard parameter sharing learns a common feature representation through shared parameters, while task-specific parameters are used to make predictions tailored to each task (Caruana, 1997). Recently, Duan & Wang (2023) proposed an augmented optimization framework for MTL that accommodates both hard parameter sharing and various types of soft parameter sharing.

We consider a general MTL objective that incorporates both parameter-sharing schemes. Specifically, consider $K$ tasks and for each task $k = 1, \ldots, K$, we observe $n_k$ independent samples, denoted by $\{z_{ki}\}_{i=1}^{n_k}$. Let $\ell_k(\cdot; \cdot)$ be the loss function for task $k$. The MTL objective is given by

$$\mathcal{L}(\boldsymbol{w}) = \sum_{k=1}^{K} \left[ \frac{1}{n_k} \sum_{i=1}^{n_k} \ell_k(\theta_k, \gamma; z_{ki}) + \Omega_k(\theta_k, \gamma) \right], \tag{1}$$

where $\boldsymbol{\theta} = \{\theta_k \in \mathbb{R}^{d_k}\}_{k=1}^{K}$ are task-specific parameters, $\gamma \in \mathbb{R}^p$ are shared parameters, $\boldsymbol{w} = \{\boldsymbol{\theta}, \gamma\}$ denotes all parameters, and $\Omega_k(\theta_k, \gamma)$ represents the task-level regularization. The parameters are estimated by minimizing Eq. (1), i.e., $\hat{\boldsymbol{w}} = \arg\min_{\boldsymbol{w}} \mathcal{L}(\boldsymbol{w})$.

Below, we present two special cases of supervised learning within this general framework: one illustrating soft parameter sharing and the other demonstrating hard parameter sharing. Let $z_{ki} = (x_{ki}, y_{ki})$ for $1 \le k \le K$ and $1 \le i \le n_k$, where $x_{ki}$ represents the features and $y_{ki}$ represents the outcomes for the $i$-th data point in task $k$.

**Example 1 (Multitask Linear Regression with Ridge Penalty).** *Regularization has been integrated in MTL to encourange similarity among task-specific parameters; see (Evgeniou & Pontil, 2004; Duan & Wang, 2023) for examples. Consider the regression setting where $y_{ki} = x_{ki}^\top \theta_k^* + \epsilon_{ki}$, with $\epsilon_{ki}$ being independent noise and $x_{ki} \in \mathbb{R}^d$ for $1 \le i \le n_k$ and $1 \le k \le K$. Additionally, we have the prior knowledge that $\{\theta_k^*\}_{k=1}^{K}$ are close to each other. Instead of fitting a separate ordinary least squares estimator for each $\theta_k$, a ridge penalty is introduced to shrink the task-specific parameters $\theta_1, \ldots, \theta_K \in \mathbb{R}^d$ toward a common vector $\gamma \in \mathbb{R}^d$, while $\gamma$ is simultaneously learned by leveraging data from all tasks.*

*The objective function for multitask linear regression with a ridge penalty is given by*

$$\mathcal{L}(\boldsymbol{w}) = \sum_{k=1}^{K} \left[ \frac{1}{n_k} \sum_{i=1}^{n_k} (y_{ki} - x_{ki}^\top \theta_k)^2 + \lambda_k \|\theta_k - \gamma\|_2^2 \right],$$

*where $\lambda_k$ controls the strength of regularization. This can be viewed as a special case of Eq. (1) by setting $\ell_k$ as the squared error (depending only on the task-specific parameters) and defining the regularization term $\Omega_k(\theta_k, \gamma) = \lambda_k \|\theta_k - \gamma\|_2^2$.*

**Example 2 (Shared-Bottom Neural Network Model).** *The shared-bottom neural network architecture, first proposed by Caruana (1997), has been widely applied to MTL across various domains (Zhou et al., 2023b; Liu et al., 2021c; Ma et al., 2018). The shared-bottom model can be represented as $f_k(x) = g(\theta_k; f(\gamma; x))$, where $f(\gamma; \cdot)$ represents the shared layers that process the input data and produce an intermediate representation, and $\gamma$ denotes the parameters shared across tasks. The function $g(\theta_k; \cdot)$ corresponds to task-specific layers, which take the intermediate representation and produce task-specific predictions, with $\theta_k$ representing task-specific parameters.*

*The loss function for this model can be written as:*

$$\mathcal{L}(\boldsymbol{w}) = \sum_{k=1}^{K} \left[ \frac{1}{n_k} \sum_{i=1}^{n_k} \ell_k(y_{ki}, g(\theta_k; f(\gamma; x_{ki}))) + \Omega_k(\theta_k, \gamma) \right],$$

where $\ell_k(\cdot, \cdot)$ *represents the task-specific loss function, and* $\Omega_k(\theta_k, \gamma)$ *denotes the regularization term. A simple choice is* $\Omega_k(\theta_k, \gamma) = \lambda_k(\|\theta_k\|_2^2 + c\|\gamma\|_2^2)$, *where* $\lambda_k$ *and* $c$ *are positive constants.*

## 3.2 Instance-Level Relatedness Measure

To quantify the instance-level contribution from one task to another, we adopt the *Leave-One-Out* (LOO)[1] principle widely used in the data attribution literature (Koh & Liang, 2017; Bae et al., 2022; Deng et al., 2024)—measuring the change in a chosen evaluation metric on the target task when a single example is omitted during training.

Formally, let $\hat{\boldsymbol{w}} = (\hat{\theta}_1, \cdots, \hat{\theta}_K, \hat{\gamma})$ denote the minimizer of Eq. (1) on the full dataset and $\hat{\boldsymbol{w}}^{(-li)} = (\hat{\theta}_1^{(-li)}, \cdots, \hat{\theta}_K^{(-li)}, \hat{\gamma}^{(-li)})$ denote the corresponding minimizer when the $i$-th data point from task $l$, i.e., $z_{li}$, is omitted. The performance of any model with parameters $\boldsymbol{w} = (\theta_1, \cdots, \theta_K, \gamma)$ on task $k$ can be measured by the average loss over a validation dataset $D_k^v$, i.e, $V_k(\theta_k, \gamma; D_k^v) = \sum_{z \in D_k^v} \ell_k(\theta_k, \gamma; z) / |D_k^v|$. The LOO effect of the $i$-th data point from task $l$ on task $k$ is defined as the difference in the validation loss when using the parameters learned from all data versus those learned by excluding the data point $z_{li}$, i.e.,

$$\Delta_k^{li} = V_k(\hat{\theta}_k, \hat{\gamma}; D_k^v) - V_k(\hat{\theta}_k^{(-li)}, \hat{\gamma}^{(-li)}; D_k^v). \tag{2}$$

Crucially, we distinguish the scope and utility of this instance-level measure from coarse-grained task-level metrics. While task-level relatedness serves as a high-level diagnostic for characterizing overall task affinity, it obscures the significant heterogeneity of data quality within each task—a phenomenon evidenced by recent studies (Lv et al., 2024; Yi et al., 2020; Zhang et al., 2023a). This oversight becomes increasingly detrimental as sample sizes grow and intra-task diversity increases. By operating at the instance level, we can pinpoint specific beneficial or harmful examples that global task-level scores would miss, enabling more precise interventions such as targeted data selection.

## 3.3 Multitask Influence Function as Efficient Approximation

Despite the fine-grained understanding of the proposed instance-level relatedness measure in Eq. (2), the computational burden of evaluating LOO effect becomes even more pronounced in MTL, particularly when the number of tasks is large. To address this computational challenge, we extend the IF-based approximation in Koh & Liang (2017) to our multitask setting. This approach builds on the idea of using infinitesimal perturbations on the weights of data points to approximate the removal of individual data points. Specifically, we introduce a weight vector $\boldsymbol{\sigma} = (\sigma_{11}, \cdots, \sigma_{1n_1}, \sigma_{21}, \cdots, \sigma_{2n_2}, \cdots, \sigma_{Kn_K}) \in \mathbb{R}^{n_1 + \cdots + n_K}$ into the MTL objective function:

$$\mathcal{L}(\boldsymbol{w}, \boldsymbol{\sigma}) = \sum_{k=1}^K \Big[ \frac{1}{n_k} \sum_{i=1}^{n_k} \sigma_{ki} \ell_{ki}(\theta_k, \gamma) + \Omega_k(\theta_k, \gamma) \Big], \tag{3}$$

where $\ell_{ki}(\cdot)$ is short for $\ell_k(\cdot; z_{ki})$. For each weight vector $\boldsymbol{\sigma}$, we denote the minimizer of Eq. (3) by $\hat{\boldsymbol{w}}(\boldsymbol{\sigma}) = (\hat{\theta}_1(\boldsymbol{\sigma}), \cdots, \hat{\theta}_K(\boldsymbol{\sigma}), \hat{\gamma}(\boldsymbol{\sigma}))$. Then the instance-level relatedness measure in Eq. (2) can be rewritten as $V_k(\hat{\theta}_k(\boldsymbol{1}), \hat{\gamma}(\boldsymbol{1}); D_k^v) - V_k(\hat{\theta}_k(\boldsymbol{1}^{(-li)}), \hat{\gamma}(\boldsymbol{1}^{(-li)}); D_k^v)$, where $\boldsymbol{1}$ is an all-ones vector and $\boldsymbol{1}^{(-li)}$ is a vector of all ones except for the $(l, i)$-th entry being 0. We approximate this difference by first-order Taylor expansion in $\boldsymbol{\sigma}$, and define the *MultiTask Influence Function* (MTIF) for the $i$-th data of task $l$ on task $k$ as:

$$\begin{aligned} \mathrm{MTIF}(i, l; k) := & \nabla_{\theta_k} V_k(\hat{\theta}_k, \hat{\gamma}; D_k^v) \cdot \frac{\partial \hat{\theta}_k(\boldsymbol{\sigma})}{\partial \sigma_{li}} \Big|_{\boldsymbol{\sigma}=\boldsymbol{1}} + \\ & \nabla_\gamma V_k(\hat{\theta}_k, \hat{\gamma}; D_k^v) \cdot \frac{\partial \hat{\gamma}(\boldsymbol{\sigma})}{\partial \sigma_{li}} \Big|_{\boldsymbol{\sigma}=\boldsymbol{1}}. \end{aligned} \tag{4}$$

Next, we derive the influence scores of the data point $z_{li}$ on the task-specific parameters $\hat{\theta}_k$ and shared parameters $\hat{\gamma}$, i.e., the partial derivatives $\partial \hat{\theta}_k / \partial \sigma_{li}$ and $\partial \hat{\gamma} / \partial \sigma_{li}$ in Eq. (4).

---

[1]While we derive the influence scores by approximating LOO, the summation of the influence scores can approximate Leave-K-Out effect under first-order approximation.

The following proposition provides explicit analytical form for the influence of a data point on task-specific parameters for the same task (within-task influence), task-specific parameters for another task (between-task influence), and shared parameters (shared influence). We first define some notation. Let $H_{kl}$ denote the $(k,l)$-th block components of the Hessian matrix of the MTL objective function $\mathcal{L}(\boldsymbol{w}, \boldsymbol{\sigma})$, as defined in Eq. (3), with respect to $\boldsymbol{w}$. This Hessian matrix has the following *block structure* in MTL:

$$H(\boldsymbol{w}, \boldsymbol{\sigma}) = \begin{pmatrix} H_{1,1} & \cdots & 0 & H_{1,K+1} \\ \vdots & \ddots & \vdots & \vdots \\ 0 & \cdots & H_{K,K} & H_{K,K+1} \\ H_{K+1,1} & \cdots & H_{K+1,K} & H_{K+1,K+1} \end{pmatrix}. \tag{5}$$

The details of each block are described in Lemma 1. We leverage the unique block structure of this Hessian in MTL to derive its analytical inverse, offering insights into how data from other tasks influence the target task through shared parameters.

**Proposition 1** (Instance-Level Within-task Influence, Between-task Influence, and Shared Influence). *Assuming the objective function $\mathcal{L}(\boldsymbol{w}, \boldsymbol{\sigma})$ in Eq. (3) is twice-differentiable and strictly convex in $\boldsymbol{w}$. For any two tasks $k \neq l$ and $1 \leq k, l \leq K$, the following hold:*

*(Shared influence) For $1 \leq i \leq n_k$, the influence of the $i$-th data point from task $k$ on the shared parameters, $\hat{\gamma}$, is given by*

$$\frac{\partial \hat{\gamma}}{\partial \sigma_{ki}} = N^{-1} \cdot H_{K+1,k} H_{kk}^{-1} \frac{\partial \ell_{ki}}{\partial \theta_k} - N^{-1} \frac{\partial \ell_{ki}}{\partial \gamma}, \tag{6}$$

*where the matrix $N := H_{K+1,K+1} - \sum_{k=1}^{K} H_{K+1,k} H_{kk}^{-1} H_{k,K+1} \in \mathbb{R}^{p \times p}$ is invertible;*

*(Within-task influence) For $1 \leq i \leq n_k$, the influence of the $i$-th data point from task $k$ on the task-specific parameters for the same task $k$, $\hat{\theta}_k$, is given by*

$$\frac{\partial \hat{\theta}_k}{\partial \sigma_{ki}} = -H_{kk}^{-1} \frac{\partial \ell_{ki}}{\partial \theta_k} - H_{kk}^{-1} H_{k,K+1} \cdot \frac{\partial \hat{\gamma}}{\partial \sigma_{ki}}; \tag{7}$$

*(Between-task influence) For $1 \leq i \leq n_l$, the influence of the $i$-th data point from task $l$ on the task-specific parameters for another task $k$, $\hat{\theta}_k$, is given by*

$$\frac{\partial \hat{\theta}_k}{\partial \sigma_{li}} = -H_{kk}^{-1} H_{k,K+1} \cdot \frac{\partial \hat{\gamma}}{\partial \sigma_{li}}. \tag{8}$$

The proof of Proposition 1 is provided in Appendix A.

**Interpretation of Instance-Level Influences.** In MTL, data points have more composite influences on task-specific parameters compared to Single-Task Learning (STL) due to interactions with other tasks and shared parameters. In STL, each data point only affects its own task's parameters through the gradient and Hessian of the task-specific objective, which is solely the first term in Eq. (7). However, in MTL, shared parameters introduce a feedback mechanism that allows data from one task to influence the parameters of other tasks. As shown in Eq. (6), the influence of $i$-th data point from task $k$ on the shared parameters stem from two sources: the first term reflects the change on the task-specific parameter $\hat{\theta}_k$, which then indirectly affects the shared parameters $\hat{\gamma}$, while the second term accounts for the direct impact on $\hat{\gamma}$. Consequently, within-task influence in Eq. (7) includes an additional influence propagated through the shared parameters, and between-task influence in Eq. (8) arises as data from one task indirectly impacts the parameters of another task via the shared parameters. In particular, in STL, between-task influence does not occur because tasks are independent and do not interact.

**Improving Computational Efficiency.** While the analytical expressions in Proposition 1 provide insight into the structure of MTIF, computing them directly requires matrix inversions involving large blocks of the full Hessian, which can be computationally expensive as the number of parameters per task or the number of tasks increases. To address this issue, numerous scalable approximations have been proposed for Hessian

inverse to improve computational efficiency, including LiSSA (Agarwal et al., 2017), EKFAC (Grosse et al., 2023), TracIn (Pruthi et al., 2020), and TRAK (Park et al., 2023). These methods approximate influence scores without explicitly computing or inverting the full Hessian. Empirically, we find that integrating the computational tricks employed by TRAK into MTIF significantly reduces the computational costs while preserving the fine-grained insight of instance-level analysis.

Specifically, TRAK (Park et al., 2023) can be viewed as a variant of influence function that incorporates several computational tricks to improve the scalability and stability of influence scores estimation, especially in the context of large neural network models. The most salient tricks used in TRAK include:

- Dimension reduction: the model parameters are projected into a lower-dimensional space using random projections to reduce computational costs.

- Ensemble: TRAK ensembles the influence scores using multiple independently trained models, which enhances the stability of the estimation against the randomness from the training process.

- Sparsification: TRAK post-processes the influence scores through soft-thresholding, which sparsifies the scores by setting the scores with small magnitudes as zero.

These tricks improves the computational efficiency and the robustness in the influence score estimation. We integrate these tricks into MTIF when applied to neural network models.

**Extension to Task-Level Relatedness.** The proposed MTIF not only provides fine-grained insight into instance-level relatedness, but also naturally extends to measure task-level relatedness. Following the same principle as LOO, we define the task-level influence of task $l$ on task $k$ using the *Leave-One-Task-Out* (LOTO) effect:

$$\Delta_k^l = V_k\big(\hat{\theta}_k, \hat{\gamma}; D_k^v\big) - V_k\big(\hat{\theta}_k^{(-l)}, \hat{\gamma}^{(-l)}; D_k^v\big), \tag{9}$$

where $(\hat{\theta}_1^{(-l)}, \cdots, \hat{\theta}_K^{(-l)}, \hat{\gamma}^{(-l)})$ is the minimizer of Eq. (3) after excluding all the data from task $l$. Analogous to instance-level MTIF, we approximate $\Delta_k^l$ by

$$\text{MTIF}_{\text{task}}(l; k) := \frac{\partial}{\partial \tilde{\sigma}_l} V_k\big(\hat{\theta}_k(\tilde{\boldsymbol{\sigma}}), \hat{\gamma}(\tilde{\boldsymbol{\sigma}}); D_k^v\big)\bigg|_{\tilde{\boldsymbol{\sigma}}=\mathbf{1}}, \tag{10}$$

where $\tilde{\boldsymbol{\sigma}} \in \mathbb{R}^K$ and

$$\hat{\boldsymbol{w}}(\tilde{\boldsymbol{\sigma}}) = \arg\min_{\boldsymbol{w}} \sum_{j=1}^K \tilde{\sigma}_j \Big[\frac{1}{n_j} \sum_{i=1}^{n_j} \ell_{ji}(\theta_j, \gamma) + \Omega_j(\theta_j, \gamma)\Big].$$

Here, $\text{MTIF}_{\text{task}}(l; k)$ captures the instantaneous change in task $k$'s validation loss when the overall weight on task $l$ is infinitesimally perturbed in the joint objective. The analytical form for $\text{MTIF}_{\text{task}}(l; k)$ is provided in Appendix B. It turns out that the task-level influence can be interpreted as a sum of instance-level influence scores over all data points in task $l$, with additional terms arising from $\sigma$-weighted regularization.

## 4 Experiments

In this section, we validate the proposed MTIF through two sets of experiments. In Section 4.1, we evaluate the quality of MTIF in terms of approximating model retraining. In Section 4.2, we further assess the practical utility of MTIF for improving MTL performance via data selection.

### 4.1 Retraining Approximation Quality

We first evaluate how well MTIF approximates the gold-standard LOO effects obtained through brute-force retraining. Given the high computational cost of repeated model retrainings needed for evaluation, we conduct this evaluation on two relatively small-scale datasets.

*Synthetic Dataset.* This dataset consists of 10 tasks, each with 200 samples $(x_{ki}, y_{ki})$ split equally into training and test sets. Inputs $x_{ki}$ are drawn independently from $\mathcal{N}(0, I_d)$ with $d = 50$, and responses are

generated using $y_{ki} = x_{ki}^\top \theta_k^\star + \epsilon_{ki}$, where $\epsilon_{ki} \sim \mathcal{N}(0, 1)$. Each task-specific coefficient $\theta_k^\star$ is obtained by perturbing a shared vector $\beta^\star = 2e_1$, where $e_1$ is the first standard basis vector. The perturbation $\delta_k$ is sampled uniformly from the sphere with radius $\|\delta\|$. We fit the soft-parameter-sharing linear MTL model described in Example 1 to estimate each $\theta_k$. Additional details are provided in Appendix C.1.1.

*HAR Dataset.* The Human Activity Recognition (HAR) dataset (Anguita et al., 2013), also referenced in Duan & Wang (2023), contains inertial sensor recordings from 30 volunteers performing daily activities while carrying a smartphone on their waist. We treat each volunteer's data as a separate binary-classification task with the objective of distinguishing the activity, "sitting", from all other activities. Preprocessing and partitioning details are provided in Appendix C.1.1. We apply the soft-parameter-sharing logistic MTL model to learn task-specific classifiers.

**Instance-Level MTIF Approximation Quality.** We compare the instance-level MTIF in Eq. (4) with the exact LOO effect in Eq. (2). The results, shown in Figure 1, reveal a strong linear correlation between the MTIF influence scores and the exact LOO scores across all scenarios. This demonstrates that MTIF effectively approximates the LOO effect for both within-task and between-task influences on the synthetic and HAR datasets.

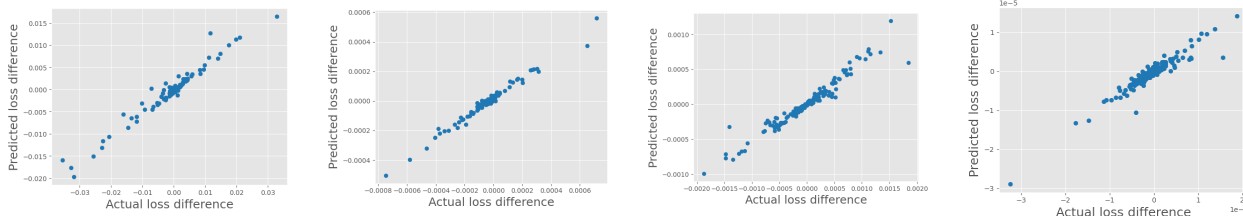

Figure 1: Instance-level MTIF approximation quality on the synthetic and HAR datasets. The x-axis is the actual loss difference obtained by LOO retraining, and the y-axis is the predicted loss difference calculated by MTIF. The first two plots from the left show within-task and between-task results (in order) results on the synthetic dataset, while the other two plots present within-task and between-task results (in order) on the HAR dataset. The plots shown here reflect influences on a randomly picked test data point, while the trend holds more broadly on other test data points. The scatter points correspond to training data points in the first task of each dataset.

**Task-Level MTIF Approximation Quality.** We compare our task-level influence scores $\text{MTIF}_{\text{task}}$ in Eq. (10) with the exact LOTO scores in Eq. (9). In each experiment, we designate one task as the target task and treat the remaining as source tasks. For each target task, we reserve 20% of its data as a validation set, compute MTIF scores for all source tasks, and obtain LOTO scores by retraining the model without each source task. We then measure the Spearman correlation between the two sets of scores, repeating the experiment for each task as the target task.

On both synthetic and HAR datasets, $\text{MTIF}_{\text{task}}$ achieves high Spearman correlation with the ground-truth LOTO scores, indicating reliable task-level relatedness estimation. Moreover, $\text{MTIF}_{\text{task}}$ outperforms two popular baselines—Cosine Similarity (Azorin et al., 2023) and TAG (Fifty et al., 2021)—in terms of correlation with LOTO. Due to page limits, we only show the results on the synthetic dataset in Table 1 and refer the readers for the results on the HAR dataset to Appendix C.1.2.

Table 1: Average Spearman correlation coefficients between $\text{MTIF}_{\text{task}}$ and LOTO scores across 5 random seeds on the synthetic dataset. Error bars represent the standard error of the mean.

| Task 1 | Task 2 | Task 3 | Task 4 | Task 5 |
|---|---|---|---|---|
| $0.84 \pm 0.05$ | $0.72 \pm 0.05$ | $0.74 \pm 0.11$ | $0.81 \pm 0.05$ | $0.71 \pm 0.09$ |
| Task 6 | Task 7 | Task 8 | Task 9 | Task 10 |
| $0.74 \pm 0.04$ | $0.74 \pm 0.07$ | $0.84 \pm 0.03$ | $0.74 \pm 0.03$ | $0.65 \pm 0.07$ |

## 4.2 Improving MTL via Data Selection

We further evaluate the utility of the proposed MTIF for improving MTL performance through data selection. While most existing MTL research focuses on task-level relatedness, the instance-level relatedness estimated by MTIF offers a unique opportunity to improve MTL performance by identifying and removing training samples that negatively impact the model.

**MTIF-Guided Data Selection.** Based on the MTL model trained on the full dataset, we first calculate the MTIF score $\text{MTIF}(i, l; k)$ for each training sample $i$ in each task $l$ with respect to each target task $k$. We then rank the training samples by their overall influence $\sum_k \text{MTIF}(i, l; k)$, and remove a fraction of the worst training samples. The removal ratio is a hyperparameter tuned on a held-out subset. Specifically, we choose the ratio that yields the highest accuracy on the validation set. Finally, we retrain the model on the selected data subset.

**Datasets.** We evaluate MTIF-guided data selection on standard MTL benchmark datasets.

*CelebA dataset.* CelebA (Liu et al., 2015) comprises over 200,000 face images annotated with 40 binary attributes and is a standard benchmark in MTL research (Fifty et al., 2021). Following Fifty et al. (2021), we select 10 attributes as separate binary classification tasks for MTL.

*Office-31 and Office-Home datasets.* Office-31 (Saenko et al., 2010) comprises three domains—Amazon, DSLR, and Webcam—each defining a 31-category classification task, with a total of 4,110 labeled images. Office-Home (Venkateswara et al., 2017) contains four domains—Artistic (Art), Clip Art, Product, and Real-World—each with 65 object categories, totaling 15,500 labeled images. Following Lin & Zhang (2023), we treat each domain as a task for MTL.

**Baseline Methods.** We compare MTIF-guided data selection against state-of-the-art MTL methods as baselines, including CAGrad (Liu et al., 2021a), Uncertainty Weighting (UW) (Kendall et al., 2018), Random Loss Weighting (RLW) (Lin et al., 2022), STCH (Lin et al., 2024), GradNorm (Chen et al., 2018b), DB-MTL (Lin et al., 2023), ExcessMTL (He et al., 2024), and PCGrad (Yu et al., 2020). The vanilla MTL model is denoted as EW (Equal Weight) following the convention in Lin & Zhang (2023). In contrast to our data-selection approach, these approaches typically mitigate negative transfer in MTL by adaptively changing gradients, task weightings, or loss scales during training. In principle, our approach can also be used in combination with these methods.

Table 2: Test accuracy of different MTL methods averaged over tasks on CelebA10, Office-Home, and Office-31 datasets. The experiments are repeated for 5 random seeds for the clean datasets and for 50 random seeds for the corrupted setups, as the random corruption introduces additional randomness. Error bars represent the standard error of the mean across the random seeds. The marks * or ** after the results of baseline methods respectively indicate that the p-value of the paired t-test between the baseline method and ours is below 0.10 or 0.05. The left three columns correspond to the original datasets while the right two columns correspond to the Office-31 dataset with respectively 10% and 20% corruption. The best result in each column is highlighted in **bold**, while the second-best result is highlighted with underline.

| Method | CelebA10 | Office-Home | Office-31 | 10% Corrupt | 20% Corrupt |
|---|---|---|---|---|---|
| EW | $79.16 \pm 0.05$ * | $78.38 \pm 0.11$ ** | $91.66 \pm 0.26$ * | $88.04 \pm 1.14$ ** | $79.37 \pm 1.79$ * |
| CAGrad | $79.46 \pm 0.08$ | $78.63 \pm 0.13$ * | $91.74 \pm 0.05$ * | $88.24 \pm 1.31$ ** | $79.52 \pm 1.17$ ** |
| UW | $79.01 \pm 0.08$ ** | $78.34 \pm 0.10$ ** | $91.87 \pm 0.16$ | $88.22 \pm 1.16$ ** | $79.20 \pm 2.00$ * |
| RLW | $79.17 \pm 0.04$ ** | $78.46 \pm 0.05$ ** | $92.00 \pm 0.13$ * | $88.66 \pm 1.30$ ** | $80.45 \pm 1.32$ |
| STCH | $79.26 \pm 0.06$ ** | $78.18 \pm 0.15$ ** | $93.19 \pm 0.08$ | $88.59 \pm 1.04$ | $78.76 \pm 1.42$ ** |
| GradNorm | $79.24 \pm 0.06$ ** | $78.55 \pm 0.12$ | $91.95 \pm 0.12$ | $88.40 \pm 1.03$ * | $79.27 \pm 1.76$ |
| DB-MTL | $79.68 \pm 0.07$ | $78.70 \pm 0.10$ ** | $93.41 \pm 0.10$ | $88.01 \pm 1.34$ ** | $79.21 \pm 0.65$ ** |
| ExcessMTL | $79.14 \pm 0.07$ ** | $78.47 \pm 0.17$ | $91.34 \pm 0.32$ | $88.87 \pm 0.99$ | $79.71 \pm 1.35$ * |
| PCGrad | $79.02 \pm 0.09$ ** | $78.33 \pm 0.08$ ** | $91.88 \pm 0.02$ * | $88.38 \pm 0.87$ ** | $79.13 \pm 1.76$ * |
| MTIF (Ours) | $\mathbf{79.94 \pm 0.04}$ | $\mathbf{79.39 \pm 0.04}$ | $\mathbf{93.60 \pm 0.01}$ | $\mathbf{89.36 \pm 1.46}$ | $\mathbf{80.97 \pm 1.28}$ |

**Experimental Setups.** We evaluate our method and the baselines in two experimental setups. The first one follows the standard MTL experimental setup using the benchmark datasets. In the second setup, we aim to highlight the heterogeneity of instance-level relatedness—specifically, that different data points from a task, rather than the entire task, may differentially affect the performance on another task. To simulate this effect, we introduce noise by randomly corrupting 10% and 20% of the labels among the training samples in the Office-31 dataset. These corrupted training samples, regardless which task they come from, are expected to be harmful to all tasks, thereby inducing heterogeneity in the instance-level influences.

**Model Training and Tuning.** For all the aforementioned datasets, we employ a pretrained ResNet-18 backbone (He et al., 2016) and attach a separate linear head for each domain's classification task. Models are trained with Adam optimizer (Kingma & Ba, 2017) with learning rate 3e-4 and weight decay 1e-5.

In all experiments we use the same validation dataset for the hyperparameter tuning and early stopping for all baselines & MTIF. The same validation dataset is also used to calculate influence scores in MTIF (so MTIF does not use extra data compared to baselines).

**Experimental Results: Accuracy.** We report the average test accuracy of different methods in Table 2. Our method (MTIF), which trains an MTL model on data selected using instance-level influence estimates, achieves the highest average accuracy across all settings, consistently outperforming all baselines. Specifically, the left three columns correspond to benchmark datasets without corruptions, while the right two columns report results on the Office-31 dataset with 10% and 20% corruptions. Compared to the original Office-31 dataset, the performance gap between MTIF and the second-best method widens as the corruption level increases. This trend indicates that MTIF is more robust to data noise and better captures fine-grained task relatedness by explicitly modeling instance-level interactions.

Table 3: End-to-end runtime (in seconds) of different MTL methods on the Office-31 dataset.

| Method | Runtime (s) |
|---|---|
| EW | $527.55 \pm 2.37$ |
| CAGrad | $1{,}121.79 \pm 2.69$ |
| UW | $592.97 \pm 2.86$ |
| RLW | $466.30 \pm 2.84$ |
| STCH | $748.15 \pm 0.01$ |
| GradNorm | $937.78 \pm 1.94$ |
| DB-MTL | $936.48 \pm 0.02$ |
| ExcessMTL | $1{,}386.13 \pm 1.62$ |
| PCGrad | $974.22 \pm 2.33$ |
| MTIF (Ours) | $1{,}281.69 \pm 0.34$ |

**Experimental Results: End-to-End Runtime.** We further compare our method with baseline MTL methods in terms of the end-to-end runtime. Our MTIF-guided data selection requires two model training passes (the initial training on the full dataset and the retraining on the selected dataset) and one evaluation pass to compute the MTIF scores. In contrast, most baseline methods perform a single model training run but incur per-step overhead to adjust gradients, task weightings, or loss scales during teh training. In Table 3, we present the end-to-end total runtime of different methods for a fair comparison. Overall, all methods exhibit comparable end-to-end runtimes, remaining within the same order of magnitude. This suggests that although MTIF-guided data selection adopts a fundamentally different approach from most existing MTL methods, its performance gains come with negligible additional computational cost.

## 5 Conclusion and Discussion

This work establishes a novel connection between data attribution and multitask learning (MTL), and introduces the MultiTask Influence Function (MTIF), a novel approach that adapts influence function-based data attribution to the MTL setting. MTIF enables fine-grained, instance-level quantification of how

individual training samples from one task affect performance on another, offering a new perspective on measuring task relatedness.

Empirically, our method achieves two key outcomes. First, we show that MTIF scores closely approximate the gold-standard leave-one-out retraining effects at both the instance and task levels. Second, we demonstrate that MTIF-guided data selection consistently improves model performance across standard MTL benchmarks, particularly in settings with heterogeneous data quality within each task, while incurring only modest additional computational overhead.

**Limitations and Future Directions.** As an initial step toward adapting data attribution methods to multitask learning, our empirical study focuses on standard computer vision MTL benchmarks. Future work could explore extending MTIF to more complex tasks and architectures, such as those involving LLMs. Additionally, influence function methods are based on first-order approximations of how infinitesimal changes in training data weights affect model performance. As a result, a potential limitation is that its task-level relatedness measure, $\text{MTIF}_{\text{task}}$—which approximates the effect of removing all data from a task—may become less accurate in approximating the LOTO effects when the number of data points per task is very large. In such cases, however, the heterogeneity within each task may be more evident, and the more fine-grained, instance-level effects may offer more meaningful insights than the LOTO effects. Better understanding the relationship and trade-offs between LOO and LOTO effects could be an interesting future direction.

# 6 Broader Impact Statement

This work introduces a data attribution–based approach for measuring task relatedness and improving multitask learning through instance-level data selection. By operating as a modular preprocessing step that does not require changes to model architectures or training pipelines, our method can be easily integrated into existing systems and may improve robustness and data efficiency in practical applications.

Our experiments include human-centric datasets such as CelebA, which contain demographic imbalances and socially sensitive attributes. Although our method does not explicitly use such attributes, influence-based data selection may unintentionally amplify existing biases if certain subgroups are disproportionately removed. Conversely, it may also help mitigate bias by reducing the impact of noisy or misleading samples. We therefore emphasize the importance of combining our approach with proper audits in scenarios with potential ethical concerns.

# Acknowledgements

Liu and Tang were supported by NSF DMS-2412853 and Jane Street Group, LLC.

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

## A Lemmas and Proofs

The first lemma describe the structure of the Hessian matrices for instance-level inference.

**Lemma 1** (Hessian Matrix Structure for Data-Level Inference). *Let $H(\boldsymbol{w}, \boldsymbol{\sigma})$ be the Hessian matrix of data-level $\boldsymbol{\sigma}$-weighted objective (3) with respect to $\boldsymbol{w}$, i.e., $H(\boldsymbol{w}, \boldsymbol{\sigma}) = \frac{\partial^2 \mathcal{L}(\boldsymbol{w}, \boldsymbol{\sigma})}{\partial \boldsymbol{w} \partial \boldsymbol{w}^\top}$, then we have*

$$
H(\boldsymbol{w}, \boldsymbol{\sigma}) = \begin{pmatrix} H_{1,1} & \cdots & 0 & H_{1,K+1} \\ \vdots & \ddots & \vdots & \vdots \\ 0 & \cdots & H_{K,K} & H_{K,K+1} \\ H_{K+1,1} & \cdots & H_{K+1,K} & H_{K+1,K+1} \end{pmatrix},
$$

*where*

$$
H_{kk} = \sum_{i=1}^{n_k} \sigma_{ki} \frac{\partial^2 \ell_{ki}(\theta_k, \gamma)}{\partial \theta_k \partial \theta_k^\top} + \frac{\partial^2 \Omega_k(\theta_k, \gamma)}{\partial \theta_k \partial \theta_k^\top},
$$

$$
H_{kl} = \boldsymbol{0},
$$

$$
H_{K+1,k}^\top = H_{k,K+1} = \sum_{i=1}^{n_k} \sigma_{ki} \frac{\partial^2 \ell_{ki}(\theta_k, \gamma)}{\partial \theta_k \partial \gamma^\top} + \frac{\partial^2 \Omega_k(\theta_k, \gamma)}{\partial \theta_k \partial \gamma^\top},
$$

$$
H_{K+1,K+1} = \sum_{k=1}^{K} \sum_{i=1}^{n_k} \sigma_{ki} \frac{\partial^2 \ell_{ki}(\theta_k, \gamma)}{\partial \gamma \partial \gamma^\top} + \sum_{k=1}^{K} \frac{\partial^2 \Omega_k(\theta_k, \gamma)}{\partial \gamma \partial \gamma^\top},
$$

*for $1 \le k, l \le K$ and $k \ne l$.*

**Lemma 2** (Influence Scores for Instance-Level Analysis). *Assume that the objective $\mathcal{L}(\boldsymbol{w}, \boldsymbol{\sigma})$ is twice differentiable and strictly convex in $\boldsymbol{w}$. Then, $\hat{\boldsymbol{w}}(\boldsymbol{\sigma}) = \arg\min_{\boldsymbol{w}} \mathcal{L}(\boldsymbol{w}, \boldsymbol{\sigma})$ satisfies $\frac{\partial \mathcal{L}(\hat{\boldsymbol{w}}(\boldsymbol{\sigma}), \boldsymbol{\sigma})}{\partial \boldsymbol{w}} = 0$. Moreover, we have:*

$$
\frac{\partial \hat{\boldsymbol{w}}(\boldsymbol{\sigma})}{\partial \sigma_{ki}} = -H(\hat{\boldsymbol{w}}(\boldsymbol{\sigma}), \boldsymbol{\sigma})^{-1} \boldsymbol{v},
$$

*where*

$$
\boldsymbol{v} = \left( 0, \cdots, 0, \underbrace{\frac{\partial \ell_{ki}}{\partial \theta_k^\top}}_{k\text{-th block}}, 0, \cdots, 0, \underbrace{\frac{\partial \ell_{ki}}{\partial \gamma^\top}}_{(K+1)\text{-th block}} \right)^\top
$$

*and $H(\boldsymbol{w}, \boldsymbol{\sigma}) \in \mathbb{R}^{(\sum_{k=1}^{K} d_k + p) \times (\sum_{k=1}^{K} d_k + p)}$ is the Hessian matrix of $\mathcal{L}(\boldsymbol{w}, \boldsymbol{\sigma})$ with respect to $\boldsymbol{w}$.*

*Proof.* The result is obtained by applying the classical influence function framework as outlined in Koh & Liang (2017). □

The following two lemmas provide tools for verifying the invertibility of the Hessian matrix and calculating its inverse.

**Lemma 3** (Invertibility of Hessian). *If $H_{kk}$ is invertible for $1 \le k \le K$, define*

$$
N := H_{K+1,K+1} - \sum_{k=1}^{K} H_{K+1,k} H_{kk}^{-1} H_{k,K+1} \in \mathbb{R}^{p \times p}. \tag{11}
$$

*If $N$ is also invertible, then $H$ is invertible.*

**Lemma 4** (Hessian Inverse). *Let $\left[ H^{-1} \right]_{k,l}$ denote the $(k,l)$ block of the inverse Hessian $H(\boldsymbol{w}, \boldsymbol{\sigma})^{-1}$. Then for $1 \le k, l \le K$,*

$$
\left[ H^{-1} \right]_{k,l} = \mathbf{1}(k = l) \cdot H_{kk}^{-1} + H_{kk}^{-1} H_{k,K+1} N^{-1} H_{K+1,l} H_{ll}^{-1},
$$

$$
\left[ H^{-1} \right]_{k,K+1} = -H_{kk}^{-1} H_{k,K+1} N^{-1},
$$

$$
\left[ H^{-1} \right]_{K+1,K+1} = N^{-1}.
$$

*Proof of Lemma 3 and Lemma 4.* Denote

$$H = \left( \begin{array}{cc} A & B \\ C & D \end{array} \right),$$

where

$$A = \left( \begin{array}{ccc} H_{11} & & 0 \\ & \ddots & \\ 0 & & H_{KK} \end{array} \right) \in \mathbb{R}^{(\sum_{k=1}^K n_k) \times (\sum_{k=1}^K n_k)}$$

$$B = C^\top = \left( \begin{array}{c} H_{1,K+1} \\ \vdots \\ H_{K,K+1} \end{array} \right) \in \mathbb{R}^{(\sum_{k=1}^K n_k) \times p},$$

$$D = H_{K+1,K+1} \in \mathbb{R}^{p \times p}.$$

Under the conditions, the matrices $H_{kk}$ for $1 \le k \le K$ are invertible. Note that $A$ is a diagonal block matrix. It is also invertible and its inverse is given by

$$A^{-1} = \left( \begin{array}{ccc} H_{11}^{-1} & & \\ & \ddots & \\ & & H_{KK}^{-1} \end{array} \right).$$

In addition, under the conditions, $D - CA^{-1}B = H_{K+1,K+1} - \sum_{k=1}^K H_{K+1,k} H_{kk}^{-1} H_{k,K+1} = N$ is invertible. Using the inverse formula for block matrix, we derive that $H^{-1}$ is

$$\left( \begin{array}{cc} \left(A - BD^{-1}C\right)^{-1} & -A^{-1}B\left(D - CA^{-1}B\right)^{-1} \\ -D^{-1}C\left(A - BD^{-1}C\right)^{-1} & \left(D - CA^{-1}B\right)^{-1} \end{array} \right), \tag{12}$$

where the upper left block is equivalent to

$$\left(A - BD^{-1}C\right)^{-1} = A^{-1} + A^{-1}B\left(D - CA^{-1}B\right)^{-1}CA^{-1},$$

by using the Woodbury matrix identity. Further, by expanding the RHS of Equation (12) in terms of the blocks in $H$, we can get the block-wise expression of $H^{-1}$. In particular, for $1 \le k, l \le K$,

$$\left[H^{-1}\right]_{k,l} \equiv \left[\left(A - BD^{-1}C\right)^{-1}\right]_{k,l}$$

$$= 1(k = l) \cdot H_{kk}^{-1} + \left[A^{-1}B\left(D - CA^{-1}B\right)^{-1}CA^{-1}\right]_{kl}$$

$$= 1(k = l) \cdot H_{kk}^{-1} + H_{kk}^{-1}H_{k,K+1} \cdot N^{-1} \cdot H_{K+1,l}H_{ll}^{-1}.$$

Further, for $1 \le k \le K$,

$$\left[H^{-1}\right]_{k,K+1} = \left[H^{-1}\right]_{K+1,k}^\top = H_{kk}^{-1}H_{k,K+1}N^{-1},$$

and

$$\left[H^{-1}\right]_{K+1,K+1} = N^{-1}.$$

$\square$

## B  MTIF for Task-Level Inference

We define task-level $\boldsymbol{\sigma}$-weighted objective to be:

$$\mathcal{L}(\boldsymbol{w}, \boldsymbol{\sigma}) = \sum_{j=1}^K \sigma_j \left[ \frac{1}{n_j} \sum_{i=1}^{n_j} \ell_{ji}(\theta_j, \gamma) + \Omega_j(\theta_j, \gamma) \right], \tag{13}$$

where $\boldsymbol{\sigma} \in \mathbb{R}^K$ is the vector of task-level $\boldsymbol{\sigma}$-weights. The first lemma describe the structure of the Hessian matrices for this task-level $\boldsymbol{\sigma}$-weighted objective.

**Lemma 1** (Hessian Matrix Structure for Task-Level Inference). *Let $H(\boldsymbol{w}, \boldsymbol{\sigma})$ be the Hessian matrix of task-level $\boldsymbol{\sigma}$-weighted objective (13) with respect to $\boldsymbol{w}$, then*

$$H(\boldsymbol{w}, \boldsymbol{\sigma}) = \begin{pmatrix} H_{1,1} & \cdots & 0 & H_{1,K+1} \\ \vdots & \ddots & \vdots & \vdots \\ 0 & \cdots & H_{K,K} & H_{K,K+1} \\ H_{K+1,1} & \cdots & H_{K+1,K} & H_{K+1,K+1} \end{pmatrix},$$

*where*

$$H_{kk} = \sigma_k \left[ \sum_{i=1}^{n_k} \frac{\partial^2 \ell_{ki}(\theta_k, \gamma)}{\partial \theta_k \partial \theta_k^\top} + \frac{\partial^2 \Omega_k(\theta_k, \gamma)}{\partial \theta_k \partial \theta_k^\top} \right],$$

$$H_{kl} = \mathbf{0},$$

$$H_{K+1,k}^\top = H_{k,K+1} = \sigma_k \left[ \sum_{i=1}^{n_k} \frac{\partial^2 \ell_{ki}(\theta_k, \gamma)}{\partial \theta_k \partial \gamma^\top} + \frac{\partial^2 \Omega_k(\theta_k, \gamma)}{\partial \theta_k \partial \gamma^\top} \right],$$

$$H_{K+1,K+1} = \sum_{k=1}^{K} \sigma_k \left[ \sum_{i=1}^{n_k} \frac{\partial^2 \ell_{ki}(\theta_k, \gamma)}{\partial \gamma \partial \gamma^\top} + \frac{\partial^2 \Omega_k(\theta_k, \gamma)}{\partial \gamma \partial \gamma^\top} \right],$$

*for $1 \leq k, l, \leq K$ and $k \neq l$.*

**Lemma 2** (Influence Scores for Task-Level Analysis). *Assume that the objective $\mathcal{L}(\boldsymbol{w}, \boldsymbol{\sigma})$ is twice differentiable and strictly convex in $\boldsymbol{w}$. Then, the optimal solution $\hat{\boldsymbol{w}}(\boldsymbol{\sigma}) = \arg\min_{\boldsymbol{w}} \mathcal{L}(\boldsymbol{w}, \boldsymbol{\sigma})$ satisfies $\frac{\partial \mathcal{L}(\hat{\boldsymbol{w}}(\boldsymbol{\sigma}), \boldsymbol{\sigma})}{\partial \boldsymbol{w}} = 0$. Furthermore, we have:*

$$\frac{\partial \hat{\boldsymbol{w}}(\boldsymbol{\sigma})}{\partial \sigma_k} = -H(\hat{\boldsymbol{w}}(\boldsymbol{\sigma}), \boldsymbol{\sigma})^{-1} \boldsymbol{v},$$

*where*

$$\boldsymbol{v} = \left( 0, \cdots, 0, \underbrace{\sum_{i=1}^{n_k} \frac{\partial \ell_{ki}}{\partial \theta_k} + \frac{\partial \Omega_k}{\partial \theta_k}}_{k\text{-th block}}, 0, \cdots, 0, \underbrace{\sum_{i=1}^{n_k} \frac{\partial \ell_{ki}}{\partial \gamma} + \frac{\partial \Omega_k}{\partial \gamma}}_{(K+1)\text{-th block}} \right)^\top$$

$H(\boldsymbol{w}, \boldsymbol{\sigma}) \in \mathbb{R}^{(\sum_{k=1}^{K} d_k + p) \times (\sum_{k=1}^{K} d_k + p)}$ *is the Hessian matrix of $\mathcal{L}(\boldsymbol{w}, \boldsymbol{\sigma})$ with respect to $\boldsymbol{w}$.*

*Proof.* The result is obtained by applying the classical influence function framework as outlined in Koh & Liang (2017). $\square$

In Proposition 2, we provide the analytical form for the influence of data from one task on the parameters of another task and the shared parameters. The Hessian matrix of $\mathcal{L}(\boldsymbol{w}, \boldsymbol{\sigma})$ with respect to $\boldsymbol{w}$ shares the same block structure as shown in (5). Let $H_{kl}$ denote the $(k, l)$-th block of the Hessian matrix, with the details provided in Lemma 1. Let $N$ be defined as in Proposition 1.

**Proposition 2** (Task-Level Between-task Influence). *Under the assumptions of Proposition 1, for any two tasks $k \neq l$ where $1 \leq k, l \leq K$, the influence of data from task $l$ on the task-specific parameters of task $k$, $\hat{\theta}_k$, is given by*

$$\frac{\partial \hat{\theta}_k}{\partial \sigma_l} = -H_{kk}^{-1} H_{k,K+1} \cdot \frac{\partial \hat{\gamma}}{\partial \sigma_l}, \tag{14}$$

*where $\frac{\partial \hat{\gamma}}{\partial \sigma_l}$ is the influence of data from task $l$ on the shared parameters, $\hat{\gamma}$, and is given by*

$$\frac{\partial \hat{\gamma}}{\partial \sigma_l} = N^{-1} H_{K+1,l} H_{ll}^{-1} \left[ \sum_{i=1}^{n_l} \frac{\partial \ell_{li}}{\partial \theta_l} + \frac{\partial \Omega_l}{\partial \theta_l} \right] - N^{-1} \left[ \sum_{i=1}^{n_l} \frac{\partial \ell_{li}}{\partial \gamma} + \frac{\partial \Omega_l}{\partial \gamma} \right]. \tag{15}$$

## C  Experiments

### C.1  Experiment Details for Retraining Approximation Quality

#### C.1.1  Synthetic and HAR Datasets and Model Configurations

**Synthetic Dataset** The synthetic dataset for multi-task linear regression is generated with $m = 10$ tasks, where each dataset contains $n = 200$ samples $(x_{ji}, y_{ji})$, split into training and test sets. The input vectors $x_{ji}$ are independently sampled from a normal distribution $\mathcal{N}(0, I_d)$ with dimensionality $d = 50$. The response $y_{ji}$ is generated using a linear model $y_{ji} = x_{ji}^\top \theta_j^\star + \epsilon_{ji}$, where $\epsilon_{ji} \sim \mathcal{N}(0, 1)$ is independent noise.

The coefficient vectors $\theta_j^\star$ for task $j$ are generated by starting with a common vector $\beta^\star = 2e_1$ (where $e_1$ is a unit vector) and adding random perturbations $\delta_j$, sampled from a sphere with norm $\delta$. For a fraction $\alpha m$ of the tasks, $\theta_j^\star$ is replaced with independent random vectors. This parameterization introduces variability in task similarity, with $\delta$ controlling the perturbation magnitude and $\alpha$ determining the fraction of unrelated tasks. For more details, we refer readers to Duan & Wang (2023).

To explore different task similarity scenarios, we generate datasets under varying $\delta$ and $\alpha$ values. The datasets are randomly divided into training, validation, and test sets with an 1:1:1 ratio.

**Human Activity Recognition (HAR) Dataset** The Human Activity Recognition (HAR) dataset (Anguita et al., 2013) was constructed from recordings of 30 volunteers performing various daily activities while carrying smartphones equipped with inertial sensors on their waist. Each participant contributed an average of 343.3 samples, ranging from 281 to 409. Each sample corresponds to one of six activities: walking, walking upstairs, walking downstairs, sitting, standing, or lying.

The feature vector for each sample is 561-dimensional, capturing information from both the time and frequency domains, and are reduced to 100 dimensions using Principal Component Analysis (PCA). To frame the dataset as a multitask learning problem, following Duan & Wang (2023), we treat each volunteer as a separate task. The problem is formulated as a multi-task logistic regression problem to classify whether a participant is sitting or engaged in any other activity. For each task, 10% of the data is randomly selected for testing, another 10% for validation, and the remaining data is used for training.

#### C.1.2  Additional Instance-Level Approximation Results

Here we present additional results for the instance-level MTIF approximation quality in Section 4.1. Figures 2 and 3 show the results on the synthetic dataset for each task selected as the target task with different $\delta$ and $\alpha$. Figures 4 and 5 show results when the data to be deleted are from different tasks than the tasks in the main text. The linear relation in both cases is still preserved, meaning our MTIF align well with LOO scores.

#### C.1.3  Additional Task-Level Approximation Results

Here we present additional results for the task-level MTIF approximation quality in Section 4.1.

**Sensitivity to synthetic data setting.** Tables 4 to 9 present results under various combinations of $\delta$ and $\alpha$. We observe that the correlation scores remain high across different settings.

**Comparison to baseline methods on more datasets and models.** We incorporate two gradient-based baselines into our task-relatedness experiments for both linear regression and neural network settings: Cosine Similarity (Azorin et al., 2023) and TAG (Fifty et al., 2021). Following the same procedure outlined in *Task-Level MTIF Approximation Quality* in Section 4.1, we evaluate task relatedness by designating one task as the target task, ranking the most influential tasks relative to it as respectively calculated by MTIF, Cosine Similarity, or TAG, and computing the ranking correlation coefficient with the ground-truth Leave-One-Task-Out (LOTO) scores. A higher correlation coefficient indicates better alignment with the LOTO scores, with values ranging from -1 (completely reversed alignment) to 1 (perfect alignment), and 0 representing random ranking. We experiment with linear regression on the synthetic dataset, logistic regression on the HAR dataset, and neural networks on the CelebA dataset.

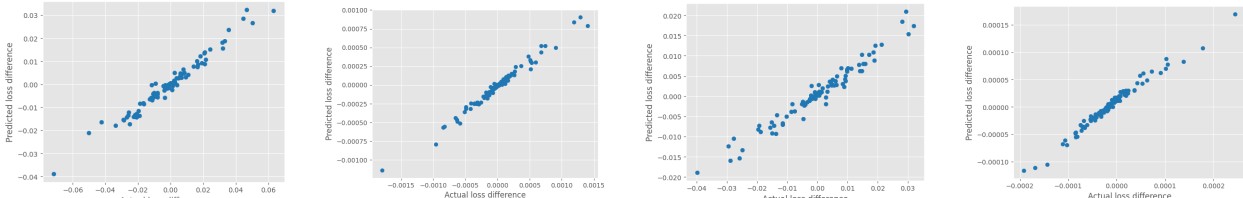

Figure 2: LOO experiments on linear regression. The x-axis is the actual loss difference obtained by LOO retraining, and the y-axis is the predicted loss difference calculated by MTIF. The first two figures from the left show within-task and between-task LOO (in order) results with $\delta = 0.4$ and $\alpha = 0$, while the other two figures present within-task and between-task results (in order) with $\delta = 0.4$ and $\alpha = 0.2$.

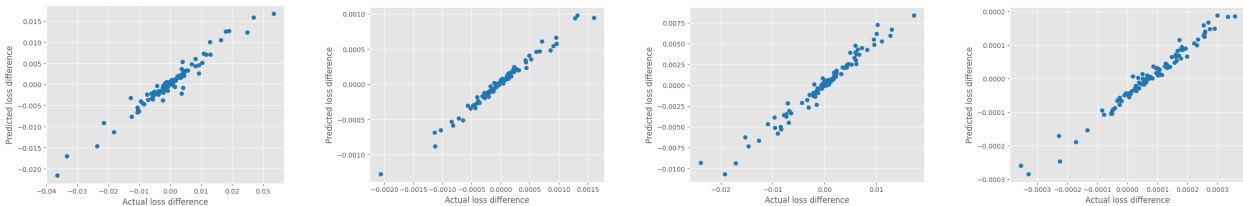

Figure 3: LOO experiments on linear regression. The x-axis is the actual loss difference obtained by LOO retraining, and the y-axis is the predicted loss difference calculated by MTIF. The first two figures from the left show within-task and between-task LOO (in order) results with $\delta = 0.8$ and $\alpha = 0$, while the other two figures present within-task and between-task results (in order) with $\delta = 0.8$ and $\alpha = 0.2$.

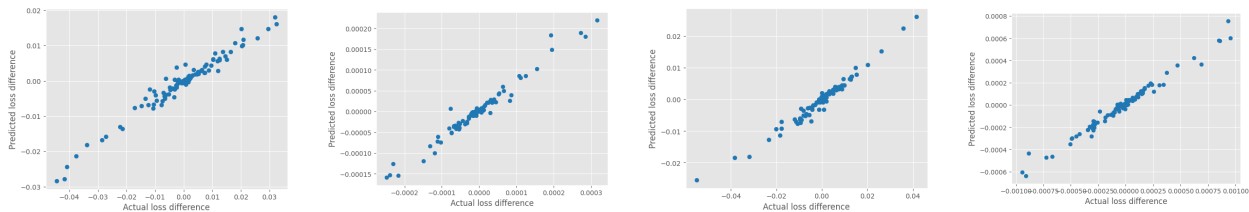

Figure 4: LOO experiments on linear regression. The x-axis is the actual loss difference obtained by LOO retraining, and the y-axis is the predicted loss difference calculated by MTIF. The first two figures from the left show within-task and between-task LOO (in order) results with deleted data from task 1, while the other two figures present within-task and between-task results (in order) with deleted data from task 2.

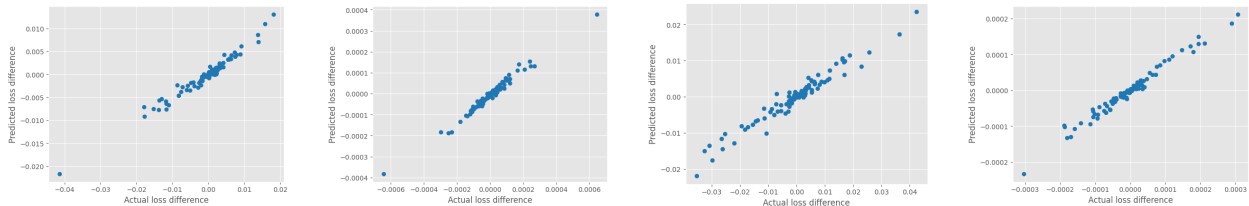

Figure 5: LOO experiments on linear regression. The x-axis is the actual loss difference obtained by LOO retraining, and the y-axis is the predicted loss difference calculated by MTIF. The first two figures from the left show within-task and between-task LOO (in order) results with deleted data from task 3, while the other two figures present within-task and between-task results (in order) with deleted data from task 5.

Table 4: The average Spearman correlation coefficients over 5 random seeds on the synthetic dataset. $\delta = 1.0$ and $\alpha = 0.2$

| Task 1 | Task 2 | Task 3 | Task 4 | Task 5 |
|---|---|---|---|---|
| $0.84 \pm 0.05$ | $0.72 \pm 0.05$ | $0.74 \pm 0.11$ | $0.81 \pm 0.05$ | $0.71 \pm 0.09$ |
| Task 6 | Task 7 | Task 8 | Task 9 | Task 10 |
| $0.74 \pm 0.04$ | $0.74 \pm 0.07$ | $0.84 \pm 0.03$ | $0.74 \pm 0.03$ | $0.65 \pm 0.07$ |

Table 5: The average Spearman correlation coefficients over 5 random seeds on the synthetic dataset. $\delta = 1.0$ and $\alpha = 0$.

| Task 1 | Task 2 | Task 3 | Task 4 | Task 5 |
|---|---|---|---|---|
| $0.75 \pm 0.07$ | $0.67 \pm 0.06$ | $0.81 \pm 0.03$ | $0.70 \pm 0.05$ | $0.60 \pm 0.10$ |
| Task 6 | Task 7 | Task 8 | Task 9 | Task 10 |
| $0.39 \pm 0.13$ | $0.66 \pm 0.06$ | $0.75 \pm 0.03$ | $0.71 \pm 0.05$ | $0.61 \pm 0.03$ |

Table 6: The average Spearman correlation coefficients over 5 random seeds on the synthetic dataset. $\delta = 0.6$ and $\alpha = 0.2$.

| Task 1 | Task 2 | Task 3 | Task 4 | Task 5 |
|---|---|---|---|---|
| $0.84 \pm 0.04$ | $0.67 \pm 0.07$ | $0.69 \pm 0.12$ | $0.77 \pm 0.05$ | $0.71 \pm 0.05$ |
| Task 6 | Task 7 | Task 8 | Task 9 | Task 10 |
| $0.73 \pm 0.07$ | $0.65 \pm 0.06$ | $0.77 \pm 0.05$ | $0.69 \pm 0.05$ | $0.56 \pm 0.11$ |

Table 7: The average Spearman correlation coefficients over 5 random seeds on the synthetic dataset. $\delta = 0.6$ and $\alpha = 0$.

| Task 1 | Task 2 | Task 3 | Task 4 | Task 5 |
|---|---|---|---|---|
| $0.77 \pm 0.05$ | $0.56 \pm 0.09$ | $0.69 \pm 0.07$ | $0.63 \pm 0.06$ | $0.57 \pm 0.13$ |
| Task 6 | Task 7 | Task 8 | Task 9 | Task 10 |
| $0.38 \pm 0.16$ | $0.62 \pm 0.04$ | $0.72 \pm 0.03$ | $0.65 \pm 0.04$ | $0.46 \pm 0.09$ |

Table 8: The average Spearman correlation coefficients over 5 random seeds on the synthetic dataset. $\delta = 0.4$ and $\alpha = 0.2$.

| Task 1 | Task 2 | Task 3 | Task 4 | Task 5 |
|---|---|---|---|---|
| $0.79 \pm 0.05$ | $0.62 \pm 0.06$ | $0.56 \pm 0.13$ | $0.73 \pm 0.05$ | $0.64 \pm 0.07$ |
| Task 6 | Task 7 | Task 8 | Task 9 | Task 10 |
| $0.67 \pm 0.08$ | $0.52 \pm 0.05$ | $0.70 \pm 0.04$ | $0.65 \pm 0.04$ | $0.56 \pm 0.09$ |

Table 9: The average Spearman correlation coefficients over 5 random seeds on the synthetic dataset. $\delta = 0.4$ and $\alpha = 0$.

| Task 1 | Task 2 | Task 3 | Task 4 | Task 5 |
|---|---|---|---|---|
| $0.67 \pm 0.08$ | $0.52 \pm 0.10$ | $0.56 \pm 0.09$ | $0.64 \pm 0.06$ | $0.54 \pm 0.15$ |
| Task 6 | Task 7 | Task 8 | Task 9 | Task 10 |
| $0.42 \pm 0.16$ | $0.52 \pm 0.08$ | $0.65 \pm 0.05$ | $0.56 \pm 0.04$ | $0.38 \pm 0.12$ |

Table 10: The average Spearman correlation coefficients over 5 random seeds on the synthetic dataset for MTIF, TAG, and Cosine across 10 tasks.

| Task | Task 1 | Task 2 | Task 3 | Task 4 | Task 5 |
|---|---|---|---|---|---|
| MTIF | $0.84 \pm 0.05$ | $0.72 \pm 0.05$ | $0.74 \pm 0.11$ | $0.81 \pm 0.05$ | $0.71 \pm 0.09$ |
| TAG | $0.57 \pm 0.03$ | $0.63 \pm 0.07$ | $0.49 \pm 0.11$ | $0.56 \pm 0.05$ | $0.69 \pm 0.04$ |
| Cosine | $0.52 \pm 0.04$ | $0.48 \pm 0.07$ | $0.39 \pm 0.12$ | $0.47 \pm 0.09$ | $0.58 \pm 0.06$ |
| Task | Task 6 | Task 7 | Task 8 | Task 9 | Task 10 |
| MTIF | $0.74 \pm 0.04$ | $0.74 \pm 0.07$ | $0.84 \pm 0.03$ | $0.74 \pm 0.03$ | $0.65 \pm 0.07$ |
| TAG | $0.55 \pm 0.12$ | $0.42 \pm 0.06$ | $0.44 \pm 0.24$ | $0.66 \pm 0.08$ | $0.61 \pm 0.07$ |
| Cosine | $0.47 \pm 0.12$ | $0.34 \pm 0.05$ | $0.40 \pm 0.22$ | $0.62 \pm 0.09$ | $0.51 \pm 0.08$ |

Table 11: The average Spearman correlation coefficients over 5 random seeds on HAR dataset for MTIF, TAG, and Cosine across 30 tasks.

|        | Task 1 | Task 2 | Task 3 | Task 4 | Task 5 | Task 6 |
|--------|--------|--------|--------|--------|--------|--------|
| MTIF   | $0.87 \pm 0.02$ | $0.90 \pm 0.02$ | $0.88 \pm 0.01$ | $0.91 \pm 0.03$ | $0.91 \pm 0.01$ | $0.90 \pm 0.02$ |
| TAG    | $0.26 \pm 0.13$ | $0.42 \pm 0.11$ | $0.55 \pm 0.09$ | $0.22 \pm 0.07$ | $0.60 \pm 0.07$ | $0.55 \pm 0.08$ |
| Cosine | $0.31 \pm 0.11$ | $0.40 \pm 0.11$ | $0.57 \pm 0.08$ | $0.20 \pm 0.09$ | $0.61 \pm 0.06$ | $0.57 \pm 0.08$ |

|        | Task 7 | Task 8 | Task 9 | Task 10 | Task 11 | Task 12 |
|--------|--------|--------|--------|---------|---------|---------|
| MTIF   | $0.90 \pm 0.01$ | $0.88 \pm 0.02$ | $0.92 \pm 0.01$ | $0.91 \pm 0.02$ | $0.89 \pm 0.02$ | $0.86 \pm 0.01$ |
| TAG    | $0.49 \pm 0.12$ | $0.31 \pm 0.12$ | $0.24 \pm 0.01$ | $0.33 \pm 0.02$ | $0.43 \pm 0.03$ | $0.21 \pm 0.02$ |
| Cosine | $0.46 \pm 0.11$ | $0.31 \pm 0.14$ | $0.26 \pm 0.03$ | $0.34 \pm 0.01$ | $0.46 \pm 0.04$ | $0.18 \pm 0.11$ |

|        | Task 13 | Task 14 | Task 15 | Task 16 | Task 17 | Task 18 |
|--------|---------|---------|---------|---------|---------|---------|
| MTIF   | $0.90 \pm 0.02$ | $0.93 \pm 0.05$ | $0.84 \pm 0.01$ | $0.87 \pm 0.05$ | $0.89 \pm 0.02$ | $0.82 \pm 0.02$ |
| TAG    | $0.54 \pm 0.03$ | $0.57 \pm 0.03$ | $0.43 \pm 0.02$ | $0.48 \pm 0.03$ | $0.64 \pm 0.05$ | $0.44 \pm 0.02$ |
| Cosine | $0.53 \pm 0.10$ | $0.58 \pm 0.10$ | $0.48 \pm 0.04$ | $0.49 \pm 0.11$ | $0.66 \pm 0.05$ | $0.46 \pm 0.07$ |

|        | Task 19 | Task 20 | Task 21 | Task 22 | Task 23 | Task 24 |
|--------|---------|---------|---------|---------|---------|---------|
| MTIF   | $0.85 \pm 0.02$ | $0.91 \pm 0.02$ | $0.93 \pm 0.02$ | $0.80 \pm 0.01$ | $0.80 \pm 0.02$ | $0.82 \pm 0.05$ |
| TAG    | $0.44 \pm 0.03$ | $0.46 \pm 0.02$ | $0.84 \pm 0.02$ | $0.52 \pm 0.07$ | $0.13 \pm 0.03$ | $0.38 \pm 0.07$ |
| Cosine | $0.48 \pm 0.05$ | $0.47 \pm 0.07$ | $0.84 \pm 0.10$ | $0.53 \pm 0.08$ | $0.16 \pm 0.12$ | $0.45 \pm 0.10$ |

|        | Task 25 | Task 26 | Task 27 | Task 28 | Task 29 | Task 30 |
|--------|---------|---------|---------|---------|---------|---------|
| MTIF   | $0.89 \pm 0.02$ | $0.81 \pm 0.03$ | $0.82 \pm 0.03$ | $0.89 \pm 0.01$ | $0.92 \pm 0.03$ | $0.86 \pm 0.03$ |
| TAG    | $0.56 \pm 0.04$ | $0.14 \pm 0.11$ | $0.41 \pm 0.10$ | $0.14 \pm 0.11$ | $0.72 \pm 0.04$ | $0.41 \pm 0.11$ |
| Cosine | $0.60 \pm 0.04$ | $0.18 \pm 0.12$ | $0.46 \pm 0.10$ | $0.15 \pm 0.10$ | $0.74 \pm 0.11$ | $0.46 \pm 0.10$ |

Table 12: The average Spearman correlation coefficients over 5 random seeds on CelebA dataset for MTIF, TAG, and Cosine across 9 tasks.

|        | Task 1 | Task 2 | Task 3 | Task 4 | Task 5 |
|--------|--------|--------|--------|--------|--------|
| MTIF   | $0.23 \pm 0.08$ | $0.44 \pm 0.19$ | $0.25 \pm 0.11$ | $0.36 \pm 0.12$ | $0.17 \pm 0.13$ |
| TAG    | $-0.10 \pm 0.13$ | $-0.10 \pm 0.14$ | $0.09 \pm 0.06$ | $0.40 \pm 0.08$ | $0.00 \pm 0.12$ |
| Cosine | $0.12 \pm 0.18$ | $0.08 \pm 0.15$ | $0.08 \pm 0.07$ | $0.37 \pm 0.08$ | $-0.10 \pm 0.13$ |

|        | Task 6 | Task 7 | Task 8 | Task 9 |
|--------|--------|--------|--------|--------|
| MTIF   | $0.35 \pm 0.08$ | $0.25 \pm 0.07$ | $0.11 \pm 0.09$ | $0.18 \pm 0.12$ |
| TAG    | $-0.42 \pm 0.08$ | $-0.26 \pm 0.17$ | $0.06 \pm 0.13$ | $0.16 \pm 0.16$ |
| Cosine | $-0.25 \pm 0.12$ | $-0.25 \pm 0.14$ | $-0.01 \pm 0.16$ | $0.05 \pm 0.12$ |

The results in Tables 10 to 12 show that our proposed MTIF method consistently outperforms the baselines across all scenarios. For the synthetic and HAR datasets, all methods achieve positive correlation scores across tasks, but MTIF consistently achieves the highest scores, often exceeding 0.7 for most tasks. In the CelebA dataset, estimating task relatedness in neural network models proves to be more challenging. While MTIF maintains positive scores, the baselines perform close to random, frequently yielding negative scores for many tasks. Although the baselines occasionally achieve slightly higher scores than MTIF on specific tasks, their performance is inconsistent. These findings underscore MTIF's reliability and superior ability to approximate task relatedness compared to the baselines.

## C.2 Additional Instance-Level Data Selection Results

To evaluate MTIF in a multi-task scene understanding setting, we additionally conduct experiments on the indoor scene understanding dataset NYUv2 (Silberman et al., 2012). In this setup, three dense prediction tasks, semantic segmentation, depth estimation, and surface normal prediction, are trained jointly. Specifically, we follow the DeepLabV3+ architecture (Chen et al., 2018a) with a dilated ResNet-50 (Yu et al., 2017) shared encoder across tasks and use an Atrous Spatial Pyramid Pooling (ASPP) module as task-specific head. Note that this backbone is larger than the ResNet-18 used in our main experiments. We similarly corrupt 20% of the training data, and apply the TRAK-based variant of MTIF for instance-level data selection on top of equal weighting (EW) and three best-performing baselines (STCH, DB_MTL, and CAGrad). The results are shown in Table 13. MTIF-based data selection (denoted "+ MTIF" in the table) improves the performance of existing multi-task learning baselines. For semantic segmentation and depth estimation, EW + MTIF achieves the strongest performance (bold) in PAcc and ranks second (bold + italic) on the remaining metrics. For surface normal prediction, MTIF provides the largest gain when combined with CAGrad. These improvements demonstrate the effectiveness of MTIF in scene understanding.

Table 13: Performance comparison of different multi-task learning methods on Segmentation, Depth Estimation, and Surface Normal Prediction. The best result in each column is shown in **bold**, and the second-best result is underlined.

| Method | Segmentation | | Depth Estimation | | Surface Normal Prediction | | | | |
|---|---|---|---|---|---|---|---|---|---|
| | mIoU↑ | PAcc↑ | AErr↓ | RErr↓ | Mean↓ | MED↓ | 11.25↑ | 22.5↑ | 30↑ |
| EW | 0.406 | 0.662 | 0.438 | 0.183 | 27.51 | 21.51 | 0.283 | 0.518 | 0.635 |
| CAGrad | 0.388 | 0.651 | 0.431 | 0.178 | 24.73 | 18.17 | 0.330 | 0.584 | 0.689 |
| UW | 0.428 | 0.678 | 0.445 | 0.184 | 27.46 | 21.55 | 0.282 | 0.517 | 0.637 |
| RLW | 0.371 | 0.629 | 0.481 | 0.192 | 28.76 | 23.72 | 0.245 | 0.478 | 0.603 |
| STCH | 0.422 | 0.673 | **0.421** | 0.178 | 25.57 | 19.22 | 0.314 | 0.563 | 0.678 |
| GradNorm | 0.432 | 0.677 | 0.433 | 0.180 | 27.23 | 21.29 | 0.287 | 0.522 | 0.640 |
| DB-MTL | 0.418 | 0.678 | 0.427 | 0.185 | 24.77 | 18.40 | 0.326 | 0.579 | 0.692 |
| ExcessMTL | 0.408 | 0.661 | 0.441 | 0.181 | 27.39 | 21.66 | 0.278 | 0.515 | 0.635 |
| PCGrad | 0.438 | 0.686 | 0.432 | 0.188 | 26.93 | 21.01 | 0.287 | 0.527 | 0.646 |
| EW+MTIF | 0.444 | **0.690** | 0.422 | 0.176 | 26.81 | 20.60 | 0.296 | 0.534 | 0.651 |
| DB-MTL+MTIF | 0.427 | 0.677 | 0.428 | 0.177 | 24.98 | 18.54 | 0.323 | 0.576 | 0.690 |
| STCH+MTIF | **0.446** | **0.690** | 0.432 | 0.177 | 25.40 | 18.82 | 0.323 | 0.569 | 0.682 |
| CAGrad+MTIF | 0.384 | 0.650 | 0.423 | **0.174** | **24.54** | **17.66** | **0.339** | **0.594** | **0.703** |

## C.3 Experiment Details for MTIF-Guided Data Selection

**Datasets.** CelebA (Liu et al., 2015) is a large-scale face image dataset annotated with 40 attributes and widely used in the multitask learning (MTL) literature (Fifty et al., 2021).

We randomly select 10 attributes as tasks for our experiments, modeling each task as a binary classification problem. The dataset is pre-partitioned into training, validation, and test sets. We sample a subset of 250 examples per task from each partition to construct our training, validation, and test sets. We do the sub-sampling as this is the regime where multitask learning outperforms single-task learning, which better

Table 14: Performance comparison of MTIF (Ours) and baseline methods on CelebA40, which treats all 40 attributes of the CelebA dataset (Liu et al., 2015) as individual tasks. Results of the baselines are reported as mean accuracy $\pm$ standard deviation over multiple runs. $^{*}$ and $^{**}$ denote paired t-test significance against MTIF with $p < 0.01$ and $p < 0.005$, respectively. See Table 2 for experimental details.

| Method | CelebA40 |
|---|---|
| EW | $84.27 \pm 0.21^{**}$ |
| CAGrad | $84.50 \pm 0.17^{*}$ |
| UW | $84.77 \pm 0.77$ |
| RLW | $84.73 \pm 0.21$ |
| STCH | $85.10 \pm 0.35$ |
| GradNorm | $84.60 \pm 0.26^{**}$ |
| DB-MTL | $85.23 \pm 0.31$ |
| ExcessMTL | $84.65 \pm 0.13^{**}$ |
| PCGrad | $84.67 \pm 0.15^{*}$ |
| MTIF (Ours) | $\mathbf{85.30 \pm 0.17}$ |

mimics the common real-world multitask learning scenarios where the training data (at least for some tasks) are scarce.

Office-31 (Saenko et al., 2010) comprises three domains—Amazon, DSLR, and Webcam—each defining a 31-category classification task, with a total of 4,110 labeled images. we partition each dataset into 60% training, 20% validation, and 20% test splits.

Office-Home (Venkateswara et al., 2017) contains four domains—Artistic (Art), Clip Art, Product, and Real-World—each with 65 object categories, totaling 15,500 labeled images. Following Lin & Zhang (2023), we treat each domain as a task for MTL. we partition each dataset into 60% training, 20% validation, and 20% test splits.

**Removal Ratio** The removal ratio is treated as a hyperparameter and selected based on validation performance (we briefly mentioned this in Section 4.2). For all reported data-selection results without data poisoning, we tune the removal ratio from {0%, 0.1%, 0.25%, 0.5%, 0.75%, 1%, 2.5%} on a held-out validation set (the same validation set used to tune hyperparameters of baseline methods) and then retrain on the remaining data. We will revise the manuscript to make this procedure clearer.

**Experimental Details** All experiments are conducted on 4 NVIDIA A40 GPUs with Linux-based system.. The following intervals of hyperparameters are explored for each method :

- **EW:** `remove_ratio` $\in [0.0, 0.005, 0.01, 0.025, 0.05, 0.1]$, `num_checkpoints` $in$ $[1, 3, 5]$.

- **CAGrad (Liu et al., 2021a):** `rescale` $\in \{0, 1, 2\}$, `calpha` $\in \{1, 2, 3\}$.

- **GradNorm (Chen et al., 2018b):** `alpha` $\in \{0.5, 1.0, 2.0\}$.

- **STCH (Lin et al., 2024):** `mu` $\in \{1.0, 2.0, 3.0\}$, `warmup_epoch` $\in \{1, 2, 3\}$.

- **DB_MTL (Lin et al., 2023):** `DB_beta` $\in \{0.5, 1.0, 2.0\}$, `DB_beta_sigma` $\in \{0.5, 1.0\}$.

- **ExcessMTL (He et al., 2024):** `robust_step_size` $\in \{0.001, 0.01, 0.1\}$.

## C.4 TRAK

When the model is small, one can compute the inverse within blocks of parameters directly. However, for large neural networks, explicitly forming and inverting the full (block) Hessian is still computationally infeasible. In those cases, we adopt the approximation tricks appeared in TRAK (Park et al., 2023) for efficient and approximate inverse Hessian. Specifically, we follow the TRAK recipe:

---

**Algorithm 1** TRAK Variant of Multi-Task Influence Functions (MTIF)

---

**Require:** Multitask learning algorithm $\mathcal{A}$ with parameters $(\gamma, \theta_1, \ldots, \theta_K) \in \mathbb{R}^{d_0 + d_1 + \cdots + d_K}$;
1: Dataset $S = \{z_{k,i} : i = 1, \ldots, n_k, \ k = 1, \ldots, K\}$ with total size $n = \sum_{k=1}^{K} n_k$;
2: Sampling fraction $\alpha \in (0, 1]$; number of subsets $M$;
3: Class-specific likelihoods $\{p_k(z; \theta_k, \gamma)\}_{k=1}^{K}$ and margins $f_k(z; \theta_k, \gamma) := \log\left(\frac{p_k(z; \theta_k, \gamma)}{1 - p_k(z; \theta_k, \gamma)}\right)$;
4: Projection dimension $d_{\text{proj}}$;
5: Validation example $z_{\text{val}}$ from task $k_{\text{val}}$;
6: Soft-threshold parameter $\lambda_S$.
**Ensure:** Attribution vector $T \in \mathbb{R}^n$ for $(z_{\text{val}}, k_{\text{val}})$.
7: **for** $m = 1$ to $M$ **do**
8:     Sample subset $S^{(m)} \subset S$ of size $\lfloor \alpha n \rfloor$
9:     Train multitask model:
$$w^{(m)} = (\theta_1^{(m)}, \ldots, \theta_K^{(m)}, \gamma^{(m)}) \leftarrow \mathcal{A}(S^{(m)})$$
10:    Sample random projection matrices:
$$P_k^{(m)} \sim \mathcal{N}(0, 1)^{d_k \times d_{\text{proj}}}, \quad k = 0, 1, \ldots, K$$
11:    Compute projected validation gradient:
$$\phi_{\text{val}}^{(m)} \leftarrow (P_{k_{\text{val}}}^{(m)})^\top \nabla_\theta f_{k_{\text{val}}}(z_{\text{val}}; \theta_{k_{\text{val}}}^{(m)}, \gamma^{(m)}) + (P_0^{(m)})^\top \nabla_\gamma f_{k_{\text{val}}}(z_{\text{val}}; \theta_{k_{\text{val}}}^{(m)}, \gamma^{(m)})$$
12:    **for** each $z_{k,i} \in S$ **do**
13:        Compute projected training gradient:
$$\phi_{k,i}^{(m)} \leftarrow (P_k^{(m)})^\top \nabla_\theta f_k(z_{k,i}; \theta_k^{(m)}, \gamma^{(m)}) + (P_0^{(m)})^\top \nabla_\gamma f_k(z_{k,i}; \theta_k^{(m)}, \gamma^{(m)})$$
14:        Compute weight:
$$q_{k,i}^{(m)} \leftarrow 1 - p_k(z_{k,i}; \theta_k^{(m)}, \gamma^{(m)})$$
15:    **end for**
16:    Stack projected gradients $\Phi^{(m)} \in \mathbb{R}^{n \times d_{\text{proj}}}$ and weights $q^{(m)} \in \mathbb{R}^n$
17:    Compute per-model influence scores:
$$t^{(m)} \leftarrow \Phi^{(m)}\left((\Phi^{(m)})^\top \Phi^{(m)}\right)^{-1} \phi_{\text{val}}^{(m)}$$
18: **end for**
19: Compute averaged attribution:
$$\bar{T} \leftarrow \left(\frac{1}{M} \sum_{m=1}^{M} q^{(m)}\right) \odot \left(\frac{1}{M} \sum_{m=1}^{M} t^{(m)}\right)$$
20: **return** $T \leftarrow \text{SoftThreshold}(\bar{T}, \lambda_S)$

---

1. linearize the model in gradient space via the task-specific margin $f_k$;

2. project per-example gradients with block-wise random projection matrices for the shared and task-specific parameters

3. estimate influence through a small linear system in the projected space, whose solution approximates the action of the inverse Hessian.

The projection dimension can also be chosen separately for each block to reflect their relative sizes. We summarize the pseudo-algorithm of this TRAK variant tailored to our MTL setting in Algorithm 1.

# D   Case Study on CelebA

In this section, we provide a case study to further make sense of the proposed method. Specifically, we identified a common pattern of negative transfer on the CelebA dataset that can be mitigated by our method. On CelebA, there are several pairs of tasks that have highly related labels, such as "Mustache" vs. "No Beard". Intuitively, these pairs of tasks are highly related and can benefit from better shared feature representation learning in multitask learning. However, while most images with a mustache also have a beard, there are also a small portion of images with a mustache but without a beard (we refer them as *semantically misaligned samples*). When the latter group of images contribute to the shared representation, it could hurt the task "No Beard", causing instance-level negative transfer. We examined the influence scores of samples in the task "Mustache" on the task "No Beard", and found that the images with a mustache but without a beard are all identified among the most negatively influencing samples. This pattern is also observed in other semantically related task pairs, including "Heavy Makeup" vs. "Wearing Lipstick" and "Wearing Necklace" vs. "Male".

Table 15 shows how the semantically misaligned samples in these task pairs are commonly identified as most negatively influencing samples. Notably, 100% of images with a mustache but without a beard are identified among the 5% most negatively influencing samples, while a dummy random influence score will give a ratio close to 5%. Similarly, 47% of images with heavy makeup but without wearing lipstick, and 32% of images wearing necklace while being male, are identified among the 5% most negatively influencing samples in their respective task pairs. These ratios are substantially higher than the expected ratio 5% from random influence scores.

This pattern also provides direct evidence that even when two tasks exhibit strong task-level relatedness, certain instances can still induce negative transfer, highlighting the value of our instance-level influence framework.

Table 15: The ratio of the number of semantically misaligned samples appearing in the top percentage of the most negatively influencing samples divided by the total number of semantically misaligned samples. The rows correspond to different task pairs. The columns correspond to the top percentage. If the influence scores are random scores, the ratio should be similar to the percentage.

| Sample semantics | 1% | 3% | 5% |
|---|---|---|---|
| With mustache and without beard | 33% | 33% | 100% |
| With heavy makeup and without wearing lipstick | 8% | 34% | 47% |
| Wearing necklace while being male | 8% | 30% | 32% |

