# OpenReview forum: "Measuring Fine-Grained Relatedness in Multitask Learning via Data Attribution"
_TMLR — Accepted by TMLR_

### Review · Reviewer_WLmJ · 2026-01-31

**Summary Of Contributions:**

This paper introduces the MultiTask Influence Function (MTIF), a method for measuring task relatedness in Multitask Learning at an instance-level scale. By adapting traditional influence functions to the MTL setting, the authors provide a way to quantify how individual training points from one task affect the performance of another without requiring exhaustive model retraining. Experiments demonstrate MTIF correlates well with the leave-one-out approach. The paper further extends this framework by applying ideas from TRAK, to support larger models.

Strengths:

S1: Solid mathematical foundations - MTIF is rooted in influence function theory and is compatible with standard MTL literature.

S2: Orthogonality - the proposed method is data-centric. Therefore, it can be combined with other MTL techniques such as GradNorm(Though see W4).

S3: Efficiency: Despite involving an attribution step, the total end-to-end runtime is comparable to existing MTL methods because it avoids the per-step overhead common in gradient-balancing techniques.

S4: Clarity: I enjoyed the writing style - clear and to the point.

Weaknesses:

W1: First-Order Approximation: As noted by the authors, MTIF relies on infinitesimal perturbations. Its accuracy for task-level influence (removing an entire task) may decrease as the number of data points per task grows very large. Similarly, the quality of the TRAK approximation may degrade as the training set grows - the local curvature at the optimum may not accurately reflect the global loss surface change when removing significant data subsets.

W2: Implementation Complexity: While scalable, implementing MTIF with TRAK-style ensembling and projections requires a more complex pipeline than standard MTL training.

W3: Limited Empirical Data: The authors conduct a number of experiments, on synthetic data. However, almost all datasets beyond the smallest ones are from the computer vision domain, and are not particularly large (e.g. OFFICE-31). Larger and more diverse datasets would provide more compelling evidence for the method’s performance, particularly in light of W1.

W4: Gaps in Literature/ Benchmarks: The paper does not mention, either as related work or as relevant benchmarks, many highly relevant approaches in transfer learning and domain adaptation, such as semi-supervised/ few-shot/ unsupervised methods. ADDA [1] and follow-up works come to mind (and can be seen as learning the relevance of data points across tasks, in a sense). How well MTIF performs in tandem with such techniques would be good to see! Works utilizing MMD [2] can also be of interest here.

[1] Adversarial Discriminative Domain Adaptation, Tzeng et al., 2017
[2] Covariate shift and local learning by distribution matching, Gretton et al.,2009

**Audience:**

Yes

**Audience Explanation:**

MTL is of great interest to many in TMLR's audience, both practically, as a tool to improve empirical performance, and theoretically, as a setting for mathematical analysis. This paper offers something to both.

**Claims And Evidence:**

Yes

**Claims Explanation:**

The math is solid, and multiple aspects of the main claims are empirically validated (correlation with LOO, benefit to training from data selection). The authors also sufficiently address concerns regarding scale and complexity.

**Requested Changes:**

The one key weakness in the list above that I believe must be addressed is a deeper discussion of the limitations as training size grows. A section expanding on the potential impact seems obligatory to me.

Others should not be blockers to the publication of a good paper, but I urge the authors to consider addressing them anyway:

The paper's positioning within transfer learning is well worth investing in - both as strong benchmarks (how does the proposed method compare with distribution-alignment approaches) and collaboratively (how may we combine MTIF with other existing techniques? What is the impact on performance?).

Verification on more diverse or large datasets will elevate the paper's potential reach, and increase the readers' confidence in the method's applicability. Moreover, it will demonstrate that the observed benefits of MTIF are not an artifact of the gradient properties specific to convolutional architectures/ computer vision.

---

> ### Author Response · Authors · 2026-02-16
>
> We thank reviewer WLmJ for the constructive feedback and suggestions!
>
> > First-Order Approximation
>
> We thank the reviewer for raising this important point. We agree that MTIF is based on a first-order approximation, and therefore its accuracy for task-level influence estimation may degrade when the number of training samples per task becomes very large. In this regime, removing an entire task corresponds to a substantial perturbation, and the local curvature around the optimum may no longer provide an accurate approximation to the global loss change. This limitation is shared by most influence-function–based and first-order methods, including approaches such as TRAK.
>
> At the same time, we would like to clarify that this regime also raises a more fundamental question about the interpretability and usefulness of task-level influence itself. When the sample size per task becomes very large, tasks often exhibit significant internal heterogeneity. In such settings, each task may contain multiple data modes or subpopulations, with mixed positive and negative influences to other tasks. Consequently, task-level influence scores may become less informative, regardless of the estimation method used.
>
> **This is closely aligned with the motivation of our work, which advocates for fine-grained influence analysis.** As training datasets grow, instance-level influence scores become more important than task-level ones. We have expanded the discussion in the revision to more clearly highlight this limitation and to clarify that task-level influence should be interpreted with caution in large-scale and heterogeneous settings.
>
> > Implementation Complexity
>
> Our method is primarily designed for data selection and does not require any modification to the model architecture or training procedure. It can be integrated into existing pipelines by inserting a lightweight data attribution step that only requires access to gradient information. As a result, compared to most existing multi-task learning methods, our approach is highly decoupled from the model training process. This modularity makes it particularly attractive for industry settings, where modifying model architectures or retraining pipelines is often costly or impractical.
>
> > Transfer Learning
>
> We agree that this is an interesting and promising direction; however, it lies beyond the scope of the present work. We plan to explore this direction in future work.
>
> > Limited empirical data.
>
> We would like to note that the Office-Home and Office-31 datasets (reported in Table 2) as well as the NYUv2 dataset (reported in Table 13 in Appendix C.2) are standard benchmarks used in recent multi-task learning literature [1, 2] (papers published in ICML 2024 and ICCV 2025), with Office-Home being the largest dataset in these papers.
> To further make sense of our proposed method, we also included a case study in the updated Appendix D. Specifically, we identified a common pattern of negative transfer on the CelebA dataset that can be mitigated by our method. On CelebA, there are several pairs of tasks that have highly related labels, such as “Mustache” vs “No Beard” or “Wearing Lipstick” vs “Heavy Makeup”. Intuitively, these pairs of tasks are highly related and can benefit from better shared feature representation learning in multitask learning. However, while most images with a mustache also have a beard, there are also a small portion of images with a mustache but without a beard. When the latter group of images contribute to the shared representation, it could hurt the task “No Beard”, causing instance-level negative transfer. We examined the influence scores of samples in the task “Mustache” on the task “No Beard”, and found that the images with a mustache but without a beard are all identified among the 5% most negatively influencing samples.
>
> This finding provides direct evidence that even when two tasks exhibit strong task-level relatedness, certain instances can still induce negative transfer, highlighting the value of our instance-level influence framework.
>
> > Additional Literature Review
>
> We thank the reviewer for highlighting the important connection to Transfer Learning and Domain Adaptation. We have updated Section 2.1 to discuss seminal works such as ADDA and MMD. We characterize these methods as **optimization strategies** that mitigate negative transfer by enforcing global feature alignment between domains. In contrast, MTIF serves as a diagnostic metric providing instance-level attribution to quantify specific data contributions. Consequently, our framework is complementary to these training methods. As the reviewer suggested, they can be effectively used in tandem—for example, using MTIF to identify and prune specific 'outlier' samples that remain harmful even after global distribution alignment.

---

> > ### Comment · Reviewer_WLmJ · 2026-02-16
> > **Response to Authors**
> >
> > I thank the authors for their thorough response and for the additions to the manuscript. The expansion of the limitations regarding first-order approximations in large-scale settings is a necessary addition for technical clarity. Furthermore, the CelebA case study (Mustache vs. No Beard) provides a very helpful intuitive grounding for the instance-level scores. I am also satisfied with the positioning of MTIF as a diagnostic tool complementary to distribution-alignment methods like ADDA and MMD. Provided these updates are fully integrated into the final version, I believe the claims are well-supported by the evidence.
> >
> > In all honesty, I would still love to see work on larger datasets. It is true that the ones in the paper are commonly used. However, not all works are equally likely to be impacted by growing complexity and size. One also notes that changes in compute and pace over the last 1-2 years may mean a rapid evolution of minimal acceptable thresholds. In any case, I am updating my recommendation to accept the paper, but urge the authors to consider "stress-testing" their approach at scale.

---

> ### Author Response · Authors · 2026-02-16
>
> > Broader Impact Statement
>
> Thanks for the suggestion. We’ve included a Broader Impact Statement at the end of the main text.
>
> [1] He, Yifei, Shiji Zhou, Guojun Zhang, Hyokun Yun, Yi Xu, Belinda Zeng, Trishul Chilimbi, and Han Zhao. "Robust Multi-Task Learning with Excess Risks." In Forty-first International Conference on Machine Learning. 2024.
>
> [2] Wang, Zedong, Siyuan Li, and Dan Xu. "Rep-MTL: Unleashing the Power of Representation-level Task Saliency for Multi-Task Learning." Proceedings of the IEEE/CVF International Conference on Computer Vision. 2025.

---

> ### Author Response · Authors · 2026-02-24
>
> Dear Reviewer WLmJ:
>
> Thank you for acknowledging that our method is well-evidenced! Regarding large-scale datasets, we are actively extending the data attribution framework for multitask learning to more scalable settings in recommender systems in an ongoing work. However, this ongoing work involves additional methodology and implementation improvement and is therefore beyond the scope of the current paper. We believe that our current framework is capable of addressing standard multi-task learning settings.
>
> Best regards,
>
> Authors of Submission 6974

---

### Review · Reviewer_P3dz · 2026-02-01

**Summary Of Contributions:**

This study introduces the Multi-Task Influence Function (MTIF) to address negative transfer in multi-task learning. Unlike traditional approaches, MTIF avoids retraining by integrating influence functions into the optimization process. The results show that MTIF matches the performance of the Leave-One-Out (LOO) method but at a fraction of the computational cost. Tested on benchmarks like CelebA and Office-31, MTIF effectively improves multi-task accuracy while maintaining high efficiency.

---

The key strengths are summarized here and weaknesses are discussed in the sections below.
- **S1. Significance of the Topic**: The paper tackles a critical challenge in Multi-Task Learning (MTL). Exploring adaptive training strategies to mitigate negative transfer remains a central and highly relevant topic for the community.
- **S2. Acknowledged Need for Efficiency**: The effort to alleviate the heavy computational burden of the conventional leave-one-out (LOO) approach, specifically the requirement for retraining, is a well-recognized practical need in the field.
- **S3. Reproducibility**: The submission includes source code, which significantly helps in verifying and reproducing the work.

**Audience:**

Yes

**Audience Explanation:**

Researchers in the Multi-Task Learning (MTL) community, especially those exploring data-centric AI and data valuation (e.g., leave-one-out influence), would be interested in seeing how traditional data selection methodologies are modified and integrated to address MTL-specific challenges like negative transfer and task-wise data conflict, potentially offering a more fundamental solution than conventional weight-balancing approaches.

**Broader Impact Concerns:**

The paper lacks a Broader Impact Statement; given the use of human-centric data like CelebA, the authors must discuss how their data-selection approach might amplify or mitigate inherent social biases (e.g., gender or race), while also reflecting on the transparency and reliability risks of applying such automated selection in high-stakes MTL fields like autonomous driving or medical diagnosis.

**Claims And Evidence:**

No

**Claims Explanation:**

Despite the aforementioned strengths, I believe the paper does not sufficiently support its claims due to the following issues:

- **W1. Lack of Justification for the LOO Foundation**: There is no theoretical or empirical evidence provided to justify why the leave-one-out (LOO) strategy is essential for this problem. The authors need to clarify why we should focus on approximating LOO performance instead of exploring alternative data valuation strategies. Furthermore, the paper lacks a discussion on existing methods for assessing the influence of data samples on specific tasks.

- **W2. Marginal Performance Gains and Scalability Concerns**: The performance improvements appear marginal, and in many cases, the gap between the proposed method and the second-best baseline does not seem statistically significant. Additionally, I am concerned about the scalability of this approach. The experiments only cover up to 10 tasks, but many MTL scenarios involve much larger scales, such as the MetaWorld MT50 track. Given the reported computation times, it is doubtful whether this method can be effectively applied to such large-scale settings.

- **W3. Limited and Outdated Literature Review**: The literature review and baseline comparisons are somewhat restricted and outdated. There are no references from 2025 and very few from 2024. Most notably, the paper misses the recent and highly relevant trend of Data Curation. I believe there are many similar strategies within that field that deserve comparison, especially since many of those methods also involve retraining, which directly relates to the LOO problem the authors aim to solve.

- **W4. Superficial Evaluation**: The experiments are somewhat shallow, consisting primarily of standard benchmark comparisons on a small number of tasks and basic timing analysis. The paper lacks a failure case analysis and does not provide an investigation into task similarity or task rejection, which are critical for understanding how data selection truly impacts MTL dynamics.

- **W5. Clarity and Presentation Issues**: The clarity of the writing needs significant improvement. There are many highly similar notations scattered throughout the manuscript, and the use of various simplified representations in later sections makes the paper difficult to follow. Due to the limited review period and the low clarity of the text, I was unable to fully verify the technical correctness of all claims.

**Requested Changes:**

- C1. Justification for LOO: Provide empirical or theoretical evidence to justify why leave-one-out (LOO) is the necessary target strategy compared to other data valuation metrics. Specifically, why we need to propose a method that can approximate the performance of LOO approaches.

- C2. Statistical Significance and Scalability: Include statistical significance tests (e.g., t-tests) for all results and provide performance/timing analysis on larger-scale benchmarks, such as MetaWorld MT50, to prove real-world scalability.

- C3. Update Literature and Baselines: Update the related work to include 2024-2025 references and compare the method against recent Data Curation or CoreSet selection strategies.

- C4. Deepen Analysis: Supplement the standard metrics with a failure case analysis and a study on how data selection impacts task similarity and task repulsion.

- C5. Improve Mathematical Clarity: Refine the notation system for better consistency and add a notation table to help the reader track the various simplified representations.

---

> ### Author Response · Authors · 2026-02-16
>
> We thank reviewer P3dz for the constructive feedback and suggestions!
>
> > Why Leave-One-Out  (LOO)
>
> We would like to first note that Leave-One-Out  (LOO) is probably the most widely adopted data attribution criterion in the literature [1, 2, 3]. Furthermore, while influence functions are derived from LOO, they could also be used to estimate Leave-K-Out. Technically, under first-order approximation, Leave-K-Out can be approximated with summation of influence functions of the data samples being left out. We have added citations and this remark in our paper to better justify the choice of LOO.
>
> > Statistical Significance and Scalability.
>
> Thanks for the suggestions. Regarding statistical significance and performance gain, we’ve included t-tests significance in Table 2 in order to show the significance of the results. In addition, we would like to highlight that the proposed MTIF is complementary to most existing MTL methods, as MTIF improves the MTL performance via instance-level data selection. In Table 13, we showed that MTIF can be used in combination with certain baseline MTL methods and the combination typically achieves better performance than the baseline method alone.
> Regarding the scalability, we would like to first note that the Office-Home and Office-31 datasets (reported in Table 2) as well as the NYUv2 dataset (reported in Table 14 in Appendix C.2) are standard benchmarks used in recent multi-task learning literature [4, 5] (papers published in ICML 2024 and ICCV 2025), with Office-Home being the largest dataset in these papers.
> We appreciate the reviewer pointing us to the MT50 dataset. However, this is a reinforcement learning dataset, while our method focuses on supervised learning. The extension of our method to reinforcement learning is non-trivial, which we leave for future exploration.
> To further demonstrate the scalability of our method with respect to the number of tasks, we have added an additional experimental section in Table 14 that includes all 40 tasks from CelebA, which is close to the 50 tasks in MT50. The results show a similar trend as the 10-task setup in our main text.
>
> > Additional Evaluation
>
> Thanks for the suggestion. We included a case study in the updated Appendix D. Specifically, we identified a common pattern of negative transfer on the CelebA dataset that can be mitigated by our method. On CelebA, there are several pairs of tasks that have highly related labels, such as “Mustache” vs “No Beard” or “Wearing Lipstick” vs “Heavy Makeup”. Intuitively, these pairs of tasks are highly related and can benefit from better shared feature representation learning in multitask learning. However, while most images with a mustache also have a beard, there are also a small portion of images with a mustache but without a beard. When the latter group of images contribute to the shared representation, it could hurt the task “No Beard”, causing instance-level negative transfer. We examined the influence scores of samples in the task “Mustache” on the task “No Beard”, and found that the images with a mustache but without a beard are all identified among the 5% most negatively influencing samples.
>
> This pattern is also observed in a few other systematically related task pairs. This finding provides direct evidence that even when two tasks exhibit strong task-level relatedness, certain instances can still induce negative transfer, highlighting the value of our instance-level influence framework.
>
> > Additional Literature Review
>
> We have updated the Related Work section to incorporate recent 2024–2025 literature on data curation and core set selection. Specifically, we acknowledge that gradient-based curation methods, such as LESS [6], share a similar foundation rooted in influence functions. However, we clarify that these approaches primarily target single-task objectives and typically lack the mechanisms to model cross-task negative transfer. MTIF extends this influence-based paradigm specifically to resolve such multitask conflicts. In parallel, we discuss other recent works that rely on alternative proxy signals including geometric boundaries, uncertainty estimates, and mixture policies, and we argue that these orthogonal directions can be effectively used in tandem with our attribution-based methodology to improve data quality.

---

> ### Author Response · Authors · 2026-02-16
>
> > Mathematical Clarity
>
> Thanks for the suggestion. We would greatly appreciate it if the reviewer could point out specific sections or examples where the presentation was particularly difficult to follow, as this would help us make more targeted revisions.
>
> As a side note, we would like to mention that another reviewer (WMmJ) commented that the paper is clear and to the point. Nonetheless, we are more than happy to further revise our paper upon receiving more concrete guidance from the reviewer.
>
> [1] Koh, Pang Wei, and Percy Liang. "Understanding black-box predictions via influence functions." International conference on machine learning. PMLR, 2017.
>
> [2] Bae, Juhan, et al. "If influence functions are the answer, then what is the question?." Advances in Neural Information Processing Systems 35 (2022): 17953-17967.
>
> [3] Deng, Junwei, et al. "$\texttt {dattri} $: A Library for Efficient Data Attribution." Advances in Neural Information Processing Systems 37 (2024): 136763-136781.
>
> [4] He, Yifei, Shiji Zhou, Guojun Zhang, Hyokun Yun, Yi Xu, Belinda Zeng, Trishul Chilimbi, and Han Zhao. "Robust Multi-Task Learning with Excess Risks." In Forty-first International Conference on Machine Learning. 2024.
>
> [5] Wang, Zedong, Siyuan Li, and Dan Xu. "Rep-MTL: Unleashing the Power of Representation-level Task Saliency for Multi-Task Learning." Proceedings of the IEEE/CVF International Conference on Computer Vision. 2025.
>
> [6] Xia, Mengzhou, Sadhika Malladi, Suchin Gururangan, Sanjeev Arora, and Danqi Chen. 2024. “LESS: Selecting Influential Data for Targeted Instruction Tuning.” In Proceedings of the 41st International Conference on Machine Learning, 235:54104–32. ICML’24 2221. JMLR.org.

---

> ### Author Response · Authors · 2026-02-24
>
> Dear Reviewer  P3dz:
>
> Thank you once again for your valuable and constructive feedback. In our response, we have significantly revised our paper, incorporating substantial additional experiments and analyses. We have also carefully addressed each review comment for every reviewer in the individual responses. We would greatly appreciate any further opportunities to engage with you in continued discussion. Thank you for your time and consideration.
>
> Best regards,
>
> Authors of Submission 6974

---

### Review · Reviewer_BXu2 · 2026-02-01

**Summary Of Contributions:**

This paper addresses a long-standing challenge in multi-task learning (MTL): measuring task relatedness and mitigating negative transfer in the presence of heterogeneous training data. The main contributions are threefold:

Multi-Task Influence Function (MTIF).
The authors extend classical influence functions to the MTL setting, covering both hard and soft parameter sharing. MTIF provides an instance-level approximation of how removing a specific training example from one task affects the validation loss of another task, enabling fine-grained cross-task influence analysis.

MTIF-guided data selection.
Based on instance-level influence scores, the paper proposes a data selection strategy that removes samples with large negative influence and retrains the MTL model on the filtered dataset. This approach aims to reduce negative transfer caused by harmful cross-task interactions.

**Additional Comments:**

No.

**Audience:**

Yes

**Audience Explanation:**

The paper should be of interest to a meaningful subset of the TMLR audience for several reasons:

Conceptual connection between MTL and data attribution.
The work bridges two active research areas—multi-task learning and influence-based data attribution—by extending influence functions to a multi-task setting. This connection is natural yet non-trivial, and may inspire further research on data-centric views of MTL.

Fine-grained analysis of task interactions.
By focusing on instance-level rather than purely task-level relatedness, the paper highlights an important and often overlooked source of negative transfer. This perspective is particularly relevant for practitioners dealing with noisy, heterogeneous, or partially misaligned multi-task datasets.

Practical and composable methodology.
The proposed data selection strategy is designed as a modular add-on that can be combined with existing MTL optimization methods. This composability increases the likelihood of adoption by researchers and practitioners working on real-world multi-task systems.

**Broader Impact Concerns:**

I do not have broader impact concerns about this paper.

**Claims And Evidence:**

Yes

**Claims Explanation:**

The core claims are supported by evidence that is largely accurate, well-aligned with the stated goals, and clearly presented. Some aspects could be further clarified, but none appear to undermine the main conclusions.

Approximation of retraining effects.
The claim that MTIF approximates retraining-based influence is convincingly supported by experiments on synthetic data and HAR, where exact LOO/LOTO retraining is feasible. The reported strong correlations between MTIF scores and retraining outcomes directly validate the approximation objective. Comparisons against baselines such as cosine similarity and TAG further strengthen this claim.

Mitigation of negative transfer via data selection.
On larger benchmarks, MTIF-guided data selection consistently improves average accuracy compared to multiple MTL baselines (e.g., PCGrad, GradNorm, CAGrad, DB-MTL, STCH). The gains are especially pronounced under label corruption, which supports the authors’ emphasis on instance-level heterogeneity as a key driver of negative transfer.

Computational feasibility.
Runtime comparisons indicate that, although MTIF introduces additional computation (influence estimation plus retraining), the overall cost remains within a comparable order of magnitude to existing approaches. The discussion of practical implementation choices (e.g., using Adachi-style approximations) helps contextualize the method’s scalability.

**Requested Changes:**

I view the following as minor revisions that would further improve clarity and reproducibility:

1. Clarify the data selection protocol in the main text.
Please explicitly describe which dataset splits are used to compute influence scores, how the removal ratio is selected, and whether the same validation set is reused for both influence estimation and model selection.

2. Emphasize the scope of instance-level vs. task-level influence.
While the limitations of task-level influence are acknowledged, it would help to more clearly state that the main empirical benefits come from instance-level MTIF, with task-level scores serving primarily as a relatedness diagnostic.

---

> ### Author Response · Authors · 2026-02-16
>
> We thank reviewer BXu2 for the constructive feedback and suggestions!
>
> > Experiment Settings
>
> Thank you for the suggestion. We clarify that for each dataset, we partition the data into training, validation, and test sets. Influence scores are computed using the training set and validation dataset, while the removal ratio is treated as a hyperparameter and selected based on performance on the validation set. The same validation set is used for both hyperparameter tuning and influence-based data selection. A description of these procedures can be found in Section 4.2 and Appendix C.2  in our original submission.
>
>
> > Scope of  instance-level vs. task-level influence
>
> We thank the reviewer for the suggestion. We agree that distinguishing the specific utility of these two measures is crucial for clarifying the scope of our contribution. Accordingly, we have added a dedicated paragraph at the end of Section 3.2 to explicitly emphasize this distinction. In this revision, we characterize task-level influence as a high-level diagnostic useful for assessing overall task affinity, while highlighting that the primary empirical benefits of our framework stem from the instance-level measure. By distinguishing these roles, we clarify that the fine-grained resolution of instance-level MTIF is what enables the precise identification and mitigation of negative transfer—capabilities that global task-level scores inherently lack due to masking data heterogeneity.

---

> ### Author Response · Authors · 2026-02-24
>
> Dear Reviewer  BXu2
>
> Thank you once again for your valuable and constructive feedback. In our response, we have significantly revised our paper, incorporating substantial additional experiments and analyses. We have also carefully addressed each review comment for every reviewer in the individual responses.  We would greatly appreciate any further opportunities to engage with you in continued discussion. Thank you for your time and consideration.
>
> Best regards,
>
> Authors of Submission 6974

---

### Decision · Action_Editor_hZQg · 2026-04-01

**Recommendation:** Accept as is

**Audience:**

Yes

**Audience Explanation:**

Yes, the paper addresses multitask learning and data attribution, which are of broad interest to the TMLR audience, particularly those working on model interpretability and transfer learning.

**Claims And Evidence:**

Yes

**Claims Explanation:**

The claims are generally supported by convincing experimental evidence, including strong correlation with LOO retraining and consistent improvements across multiple benchmarks. However, further validation on more diverse tasks would strengthen the conclusions.